

# Anthropogenic VOC in Abidjan, southern West Africa: from source quantification to atmospheric impacts

Pamela Dominutti[1*], Sekou Keita [2,3], Julien Bahino[2,3], Aurélie Colomb[1], Cathy Liousse[2], Veronique Yoboué[3], Corinne Galy-Lacaux[2], Laëtitia Bouvier[1], Stéphane Sauvage[4] and Agnès Borbon[1]

[1] Laboratoire de Météorologie Physique LaMP-OPGC-CNRS, Université Clermont Auvergne, Clermont-Ferrand, France

[2] Laboratoire d'Aérologie, Université Paul Sabatier Toulouse 3 - CNRS, Toulouse, France

[3] Laboratoire de Physique de l'Atmosphère (LAPA)- Université Felix Houphouët-Boigny, Abidjan, Côte d'Ivoire

[4] IMT Lille Douai, Sciences de l'Atmosphère et Génie de l'Environnement (SAGE), Douai, France

*Now at Wolfson Atmospheric Chemistry Laboratories, Department of Chemistry, University of York, Heslington, York, YO10 5DD, UK

Correspondence to P. Dominutti (pamela.dominutti@york.ac.uk) + A. Borbon (agnes.borbon@uca.fr )

## Abstract

Several field campaigns were deployed in the framework of the project Dynamics-Aerosol-Chemistry-Cloud Interactions in West Africa (DACCIWA) to measure a broad range of atmospheric constituents. Here, we present the analysis of an unprecedented and comprehensive dataset integrating up to fifty-six VOC from ambient sites and emission sources. VOCs were collected on sorbent tubes in the coastal city of Abidjan, Côte d'Ivoire, in winter and summer 2016 and analysed by gas chromatography coupled with flame ionization and mass spectrometer detectors (GC-FID and GC-MS) at the laboratory.

The comparison between VOC emission source profiles and ambient profiles suggests the substantial impact of two-wheelers and domestic fires on the composition of Abidjan's atmosphere. However, despite the high VOC concentrations near-source, moderate ambient levels were observed (by a factor of 10 to 4000 lower) and similar to the ones observed in northern mid-latitude urban areas. Beyond photochemistry, reported high wind speeds suggest that meteorology is an essential factor that regulates air pollution in Abidjan.

Emission ratios ($\Delta$VOC/$\Delta$CO) were established based on real-world measurements achieved on a selected number of representative combustion sources. Maximal molar mass contributions were observed from two-wheeler (TW) emissions, overpassing other regional sources by two orders of magnitude. This source also largely governs the VOC atmospheric impacts in terms of OH reactivity, secondary aerosol formation (SOAP) and ozone production (POCP). While the contribution of aromatics dominates the atmospheric impact, our measurements reveal the systematic presence of anthropogenic terpenoids in all residential combustion sectors. Finally, emission factors were used to retrieve and quantify VOC emissions from the main anthropogenic source sectors at national level. Our detailed VOC emissions estimation suggests that the road transport sector is the dominant source in Cote d'Ivoire by emitting



around 1200 Gg yr$^{-1}$ of gas-phase VOCs. These new estimations are 100 and 160 times larger than previous global inventory estimations like MACCity or Edgar (v4.3.2). Additionally, the residential sector is also largely underestimated by a factor of 13 to 43. For the only Cote d'Ivoire, these new estimates for VOCs are three to six times higher than the whole Europe. Given the significant underestimation of VOC emissions from transport and residential sectors for Côte d'Ivoire country there is an urgent need for such an effort in the whole West Africa region for building more realistic and region-specific emission inventories. This is not only true for VOCs but all atmospheric pollutants. The dearth of waste burning, fuelwood burning and charcoal representation in regional inventories need also to be addressed, particularly in low-income areas where these types of activities are essential sources of VOC emissions.

**Keywords:** VOCs, emission inventories, West Africa, air pollution, emission ratios.

## 1. Introduction

Western Africa region, located in the north of the Gulf of Guinea, is one of the most populated areas in Africa with more than 300 million inhabitants in 2016 (United Nations, 2017). The increase in population by a factor of five since 1950, converted West Africa as the fastest growing region of any of the world's regions. Furthermore, future projections indicate an increase in the population density of developing countries, which will be much higher in Africa than in other parts of the world. The population of Africa could represent 40% of the world's population in 2100 (United Nations, 2017). The explosive unplanned growth of urban conglomerations is the main issue in the region, with water access, air pollution, health problems and unregulated emissions pointed out as the principal shortcomings.

Consequently, these emissions can produce diverse effects on atmospheric chemistry enlarged by severe photochemical conditions and dynamical atmospheric interactions. The atmospheric composition over West Africa is affected by air masses transported from remote sources, like aerosol dust from the Sahara desert or biomass burning aerosol and local urban pollution (Knippertz et al., 2017; Mari et al., 2011). Observations performed during the AMMA (African Monsoon Multidisciplinary Analyses) campaign have shown that air quality issues are mainly related to traffic and combustion emissions (Mari et al., 2011). Residential emissions in Southern West Africa (SWA) are attributed to charcoal and wood burning as the primary source of domestic energy, widely used for cooking and heating activities. Biomass burning is a significant source of carbonaceous aerosols and volatile organic compounds that can have effects on public health and climate through the formation of secondary pollutants (Gilman et al., 2015; Knippertz et al., 2015; Sommers et al., 2014).

Additionally, in most SWA cities, traffic emissions are essential sources of air pollution (Assamoi and Liousse, 2010). The road transport sector is largely disorganized due to the underdevelopment of road networks and to the absence of a regulation policy for public transport (Assamoi and Liousse, 2010). In this context, two-wheeler vehicles are widely used in the cities for short-distance and transport, replacing the public transport modes. Furthermore, the vehicular fleet has increased in the last year, which is characterized, in most cities, by a large number of old vehicles (Keita et al., 2018).

In the next years, African emissions from the combustion of fossil fuels, biofuels and refuse are predicted to increase considerably and could represent about 50% of the global emissions of organic carbon (Knippertz et al., 2017; Liousse





et al., 2014). Nevertheless, detailed emissions are still required for a better estimating of their impacts on climate change and health over this highly sensitive region (Knippertz et al., 2017).

Volatile organic compounds (VOCs) include a large number of species which can affect air quality by producing secondary pollutants such as ozone and secondary organic aerosols (Seinfeld and Pandis, 2006). Given the reactive nature of VOCs (Atkinson and Arey, 2003a), their emissions need to be disaggregated on species or species groups, for a better representation of their chemical features and thus, their impacts on the secondary formation processes. VOCs are important pollutants present in the urban atmospheres; therefore in-situ VOC observations are necessary to provide direct implications on air pollution exposure and to improve the prediction of secondary products formation.

Several field campaigns have been developed in the last twenty years all over the world with the purpose to identify VOC species and their fate in the atmosphere, as well as to quantify their emission sources (Bechara et al., 2010; Bon et al., 2011; Borbon et al., 2013; Brito et al., 2015; Dominutti et al., 2016; Kumar et al., 2018; Salameh et al., 2015; Wang et al., 2014; Warneke et al., 2016). VOCs field observations have also been intensely used as constraints for the development of reliable emission inventories (Borbon et al., 2013; Salameh et al., 2016b). Some studies pointed out significant discrepancies between inventory estimations and emission ratios derived from ambient measurements, implying some limitations in the accurate modelling of VOCs impact. In addition, there are still missing observations of VOCs in sensitive places with high anthropogenic pressure like Africa and South America (Huang et al., 2017). Particularly for Africa, the emission inventories frequently used are those developed for global scales and it involves numerous uncertainties (Keita et al., 2018; Liousse et al., 2014). The main differences are associated with the emission source estimations, source categories, and spatiotemporal characterization. For VOCs, discrepancies up to a factor of 10, were observed in the comparison of emission inventories to the observations in Northern mid-latitude cities (Borbon et al., 2013; Boynard et al., 2014).

Furthermore, global emission inventories commonly reported the total mass of VOCs. Nonetheless, the fate and contribution of each species can change depending on the emission source, fuel quality, combustion technologies, and main regional activities (Huang et al., 2017). Therefore, the use of local activity data and emission factor derived from local measurements on specific sources may help reducing uncertainties in emission inventories. A recent study calculated the emission factors (EFs) of different compounds and activities in SWA (Keita et al., 2018). A comparison with the emission calculated from the EFs with those observed from EDGARv4.3.2 (Huang et al., 2017) inventory showed a marked discrepancy, by a factor of 50, for Côte d'Ivoire. This work has demonstrated the urgency to consider African anthropogenic emissions at regional scales. African anthropogenic emissions require more detailed studies to reduce the uncertainties of emission estimation which currently cannot be assessed due to the scarcity of suitable data. Therefore, the characterization and quantification of these emissions is a crucial input to improve the comprehension of anthropogenic and natural sources contribution in the atmospheric composition over SWA and to assess their impact on human and air quality conditions.

Several intensive field campaigns in the framework of the Dynamics-aerosol-chemistry-cloud-interactions in West Africa (DACCIWA) project were deployed in 2015 and 2016. Here, we present the results obtained from the VOC field campaigns at different sites in one of the major SWA city: Abidjan in Côte d'Ivoire, which included ambient and near-source measurements. Speciated VOCs were collected off-line by using sorbent tubes and then analysed and quantified at the laboratory by applying different gas chromatography techniques. These data provide the first



constraints for the construction of regional emission inventory and the understanding of the role of VOC anthropogenic emissions in the regional atmospheric chemistry.

This work aims to establish VOC speciated profiles of primary anthropogenic sources in Abidjan by performing sampling close to emission sources in real-condition operation as far as possible. These sources include traditional
and regional-specific ones, such us road transportation (gasoline, diesel, and two-wheelers emissions), charcoal fabrication and burning emissions from domestic cooking fires, charcoal, landfill waste, and fuel hardwood. A critical analysis of the profiles is also provided by comparing to ambient measurements performed at nine different sites in Abidjan. VOC ambient concentrations are also compared with those from other cities worldwide to discuss discrepancies and commonalities. The dataset obtained in our study provide substantial information enabling the
quantification of VOC emissions for several sources in Cote d'Ivoire. These sources profiles are analysed here and contrasted with those provided by global emission inventories. Finally, the impact due to the use of region-specific sources is tackled regarding the implication on air quality in terms of reactivity and secondary pollutants formation.

## 2. Materials and Methods

As part of the DACCIWA project (Knippertz et al., 2015) intensive field campaigns were performed in 2015 and 2016, focusing for the first time on the most populated southern coastal region of West Africa. The DACCIWA campaign had an emphasis on atmospheric composition, including questions of air pollution, health impacts and cloud-aerosol interactions (Knippertz et al., 2015). Here we present the results obtained from intensive ambient measurements in Abidjan and the extended VOC speciation acquired from source emission measurements (Keita et
al., 2018). These results are part of the activities that were developed under the work package 2 (WP2): Air pollution and Heath, which aims to link and quantify emission sources, air pollution and related health impacts over differentiated urban sources in West Africa (https://www.dacciwa.eu).

### 2.1 Sampling

The field campaigns were conducted in Abidjan, Côte d'Ivoire during summer and winter time according to the strategic directions of the DACCIWA WP2. Firstly, direct source emission measurements were performed to obtain emission profiles from the main sources in Abidjan, SWA. They integrated a wide range of human activities susceptible to VOCs emissions, as well as other gas-phase and particulate pollutants. The assessed sources combine traditional ones like road transportation and regional-specific of SWA such as domestic waste-burning, charcoal
fabrication, charcoal-burning as well as fuel wood-burning (**Table1**). Part of these results for a selected number of VOC (fifteen species), BC and particles were already discussed in a recently published paper (Keita et al., 2018).

- For the road transportation analysis of different vehicle exhausts tailpipe measurements were carried out. Samples integrate heavy-duty diesel vehicles (HDDV, trucks, and buses), light-duty diesel vehicles (LDDV, diesel cars), light-duty gasoline vehicles (LDGV, gasoline cars), two-wheelers 2 strokes (TW 2T) and two-
wheelers 4 strokes (TW 4T) vehicles. Differences in motorizations (gasoline and diesel) and the fleet age have been considered.

- Regarding domestic waste-burning (WB), samples were obtained at the official domestic landfill site located at the east of Abidjan (AD, Figure 1 and Table 2). The sampling was performed inside the waste burning plume to integrate the different combustion processes involved.



–   Charcoal-burning (CH) and fuel wood-burning (FW) are commonly cooking and heating practices in African urban areas. FW emissions were obtained by measuring the fire plume of tropical African hardwood, specifically Hevea (*Hevea brasiliensis*). FW and CH were burned in two types of stoves traditionally used in the SWA region for cooking, which are made of metal and baked earth. The measurements included all the combustion phases (Keita et al., 2018).

–   The charcoal-making (CHM) profile was obtained by measuring emissions from traditional kilns, that use different types of dense wood. The kiln was covered with a layer of leaves and a layer of soil of about 10 cm thick. The smoke was sampled at the holes made in the CHM kiln, which are located in the horizontal plane and provide the air circulation for the pyrolysis propagation ( Keita et al., 2018).

All samples were obtained in the emission plume at around 1–1.5m from the source, except for vehicles where

samples were taken at the tailpipe outlet while the vehicle's engine was idling. The EF values estimated from different samples were later averaged for every source and each VOC species. For the road transport sector, the equivalent fleet means were calculated considering the fleet characteristics in Côte d'Ivoire, as detailed in Keita et al.(2018). These calculations were based on the information given by the Direction Generale des Transports Terrestres in Cote d'Ivoire, which considered that 60% of vehicles are old models and 77% of the total fleet is composed by light-duty

vehicles. Regarding TW, 60% of them are two-stroke engines and only 40% of the total are considered as recent vehicles (SIE CI, 2010).

Secondly, VOC ambient measurements were collected to analyse the spatial distribution of these compounds from different sites in Abidjan, Cote d'Ivoire. Ambient measurements were performed at nine different sites, which are shown in Figure 1. The distribution of the sampling points was selected to cover the primary sources and activities

developed in the city. The selection was also in agreement with the near-source measurements, previously described. Therefore, residential, road transport, domestic fires, waste burning, industrial and background sites were selected. The characteristics and geographical location of each site are reported in Table 2. The ambient campaigns were conducted for 1 month during the dry season (February 2016) and samples were collected once a week at different daytime. An active sampling of VOCs using a manual pump (Accuro 2000, Dräger) was carried out on sorbent

cartridges due to their easily handle and to the possibility to obtain profiles from a large spatial area. Each tube was exposed several times a week at each site which corresponds to a total volume of 600 mL. Cartridges deployed for VOC measurements were composed of Tenax TA 60-80 mesh (250 mg) or Carbopack C (150 mg). The combination of both sorbent material allowed the sampling of 10 aromatics (C6-C9), 22 n-alkanes (C5-C16), 10 monoterpenes, 7 aldehydes, isoprene, and other oxygenated compounds. All compounds are reported in Table SM1. Before the

sampling, multi-sorbent filled cartridges were conditioned by flowing purified nitrogen through them, at a flow of 100 mL.min$^{-1}$, during 5 hours at 320 °C.

*2.2 Analytical instrumentation*

Duplicate measurements were performed and lately analysed in two different laboratories to investigate the

reproducibility of analytical techniques and with the purpose to acquire a wider range of VOC species. The analysis of the Tenax TA tubes was performed at the *Laboratoire de Météorologie Physique* (LaMP, Clermont-Ferrand, France) by using a gas chromatograph mass spectrometer system (GC/MS, Turbomass Clarus 600, Perkin Elmer®) coupled to automatic thermal desorption (Turbomatrix ATD). Each tube was desorbed at 270 °C during 15 min at a



flow rate of 40 mL. min⁻¹ and pre-concentrated on a second trap, at -30°C containing Tenax TA. After the
cryofocusing, the trap was rapidly heated to 300°C (40°. s⁻¹) and the target compounds were flushed into the GC.
Due to the high loads in some samples, an inlet and outlet split of 5 mL.min⁻¹ and 2 mL min⁻¹ Were set up,
respectively. The analytical column was a PE-5MS (5% phenyl – 95% PDMS, 60m×0.25mm×0.25μm) capillary
column (Perkin Elmer) and a temperature ramp was applied to guarantee the VOCs separation (35°C for 5 minutes,
heating at 8°C min⁻¹ to 250°C, hold for 2 minutes). The mass spectrometer was operated in a Total Ion Current (TIC)
from 35 to 350 m/z amu. Chromatography parameters were optimized to enable good separation of fifteen identified
compounds by a complete run of 34 minutes on each cartridge. Calibration was performed by analysing conditioned
cartridges doped with known masses of each compound, present in certified standard low-ppb gaseous standard,
purchased from the National Physical Laboratory (NPL, UK). The cartridges were then analysed with the method
described above and calibration curves were obtained for each compound as already discussed in Keita et al., (2018).
The method provided the separation and identification of 16 compounds, from C5 to C10 VOCs, including 8
aromatics, 3 monoterpenes, 4 alkanes and isoprene.

Carbopack tubes analysis were carried out by applying a gas chromatography-flame ionization detector (ATD-GC-
FID, Perkin Elmer) system at the *SAGE Department (IMT Lille Douai)*. The cartridges were previously thermo-
desorbed at 350 °C during 15 minutes with a helium flow of 20 mL.min⁻¹. This method allowed the separation and
identification of up to 56 compounds, from C5–C16 VOCs, including 7 carbonyls, 4 ketones, 10 monoterpenes and
6 VOCs of intermediate volatility. More details of this technique can be found else-where (Ait-Helal et al., 2014;
Detournay et al., 2011). The application of both methods allowed the comparison of common compounds that were
measured in ambient sites and sources (benzene, toluene, ethylbenzene, *m+p*-xylene, *o*-xylene, trimethylbenzenes,
n-heptane, iso-octane, n-octane, α-pinene, β-pinene, limonene, isoprene) and the analysis of analytical techniques'
performance. Likewise, the combination of different sorbent tubes and analytical strategies allowed the quantification
of a higher number of VOC species, and therefore an extensive analysis of source contributions. The uncertainties
and quality control parameters for both analytical methods can be found else-were (Detournay et al., 2011; Keita et
al., 2018)

*2.3 Metrics and calculations*

Different calculations were implemented to assess the VOC emissions and their impacts in Abidjan. Here, we provide
the equation basis for each parameter investigated. Firstly, the emission factors (EF) which were previously obtained
for a restricted number of VOC species (Keita et al., 2018), were also computed for the whole comprehensive VOC
database (56 compounds). EF were later used for the estimation of VOC emissions in Cote d'Ivoire for each target
source, further described in section 3.5. Secondly, the emission ratios (ER) of each VOC species related to CO for
all the emission sources were established. Finally, the reported ER were subsequently used to evaluate the
implications on atmospheric reactivity by applying commonly used metrics.

2.3.1 Emission factors

VOC emission factors were estimated from the concentrations measured for all the emission sources, as follows

$$EF(VOC) = \frac{\frac{\Delta VOC}{\Delta CO + \Delta CO_2} \, x \, MW_{VOC}}{12} \, x \, fc \, x \, 10^3 \qquad (1)$$



where EF (VOC) is the emission factor of the specific VOC in gram per kilogram of burned fuel (g kg$^{-1}$); $\Delta VOC =$ [VOC]emission − [VOC]background is the VOC mixing ratio in the emission and background air, respectively (ppbv); MW$_{VOC}$ is the molar weight of the specific VOC (in g mol$^{-1}$), 12 is the molar weight of carbon (g mol$^{-1}$) and

*fc* is the mass fraction of carbon in the fuel analysed. The *fc* values used were obtained from the literature and applied to each source. The EF for fifteen VOCs were already published and more details about the method can be found elsewhere (Keita et al.; 2018). Here, we applied the same method for the whole VOC database.

### 2.3.2 Emission ratios

Emission ratios (ER) were obtained for each VOC compound related to carbon monoxide (CO), and there were calculated as follows:

$$ER = \frac{[VOC]\ ppbv}{[\Delta CO]\ ppmv} \qquad (2)$$

We selected CO as a combustion tracer because most of VOCs and CO are co-emitted by the target sources. Furthermore, ratios to CO are regularly reported in the literature for biomass burning and urban emissions (Baker et

al., 2008; Borbon et al., 2013; Brito et al., 2015; Gilman et al., 2015; de Gouw et al., 2017; Koss et al., 2018; Wang et al., 2014), which can be useful for further comparisons. Emission ratios were calculated in ppb$_v$ of VOC per ppm$_v$ of CO, which is similar to a molar ratio (mmol VOC per mol CO). Molar mass (MM) emissions ratios were also computed. MM is the VOC mass emitted (μg.m$^{-3}$) per ppm$_v$ CO, obtained from equation 2 and converted by using the VOC molecular weight (g mol$^{-1}$) and the molar volume (24.86 L at 1 atm and 30 ∘C). Table SM1 includes the

emission ratios obtained for each VOC and MW values used.

### 2.3.3 VOC-OH reactivity

The OH reactivity was estimated to evaluate the contribution of each VOC measured and to identify key VOC reactive species that should contribute to the photochemical processing over SWA. VOC-OH reactivity represents

the sink reaction of each VOC with the hydroxyl radical (OH) and is equal to

$$VOC_{OH\ reactivity} = ER\ x\ kOH\ x\ CF\ , \qquad (3)$$

where ER is the emission ratio for each VOC related to CO (ppb$_v$ per ppm$_v$), *k*OH is the second-order reaction rate coefficient of VOC with the hydroxyl radical (x10$^{-12}$ cm$^3$ molec$^{-1}$ s$^{-1}$) and CF is the conversion factor of molar

concentration (2.46×10$^{10}$ molec cm$^{-3}$ ppb$_v$$^{-1}$ at 1 atm and 25 ∘C) (Gilman et al., 2015). *k*OH values for all VOC species were obtained from Atkinson and Arey (2003a) and the NIST Chemical Kinetics Database (Manion et al., 2015).

### 2.3.4 Ozone Formation Potential

The oxidation of VOCs is often initiated by reaction with the hydroxyl radical (·OH), which in the presence of NO$_x$

(NO+NO$_2$) leads to the photochemical formation of O$_3$. The ozone formation potential represents the ability of each VOC to produce tropospheric ozone and it was calculated as follows

$$VOC - ozone\ formation\ potential = ER\ x\ POCP, \qquad (4)$$

where the ER is the emission ratio of each VOC related to CO (ppb$_v$ of VOC per ppm$_v$ of CO) and POCP is the photochemical ozone creation potentials developed by Derwent et al. (Derwent et al., 2007, 2010a; Jenkin et al.,



2017). POCP values were obtained by deploying the Master Chemical Mechanism (MCM), a highly detailed chemical mechanism, applied on a realistic air mass trajectory from urban sites. This model estimates the change in ozone production by incrementing the mass emission of each VOC (Derwent et al., 1998). POCPs for an individual VOC is estimated by quantifying the effect of a small increase in its emission on the concentration of the modelled ozone formed, respective to that resulting from the same increase in the emission of ethene (POCP value for ethene

is therefore 100). In this study POCP values were also analysed on VOC family basis, which was obtained from a recent study (Huang et al., 2017) or adapted from individual POCP values.

2.3.5 SOA formation potential

The secondary organic aerosol formation potential represents the propensity of each VOC to form organic aerosols

and is equal to

$$SOA - VOC\ formation\ potential = ER\ x\ SOAP\ , \hspace{2cm} (5)$$

where ER is the emission ratio for each measured VOC related to CO ($ppb_v$ of VOC per $ppm_v$ of CO) and SOAP is a non-dimensional model-derived SOA formation potential (Derwent et al., 2010b; Gilman et al., 2015). All SOAP values represent the modelled mass of organic aerosol formed per mass of VOC reacted on an equal mass emitted

basis relative to toluene. Toluene was selected as the reference compound due to its well-known emissions and it is usually documented as a critical anthropogenic SOA precursor (Derwent et al., 2010b).

ER, $k$OH, SOAP and POCP values for each VOC are detailed in Table SM1, In the absence of those values for specific compounds, we estimated the values (indicated in Table SM1 by a +) by using those of comparable

compounds based on similar chemical properties, as suggested in the study of Gilman et al. (2015).

*2.4 Ancillary data*

Meteorological observations were provided by the NOAA Integrated Surface Database (ISD; https://www.ncdc.noaa.gov/isd for more details). Daily rainfall, air temperature, and wind speed and direction measurements

were recorded at the Abidjan Felix Houphouet Boigny Airport. Figure 1 gives the coordinates location of the meteorological station and ambient sampling sites.

## 3. Results and discussion

*3.1 Local and synoptic meteorological conditions*

Abidjan is the economical capital of Côte d'Ivoire with a population of 6.5 million (in 2016), representing more than 20 per cent of the overall population of the country (United Nations, 2017). Abidjan together with the autonomous district encompasses an area of 2,119 $km^2$ and is distinguished by remarkable industrialization and urbanization. In summer, West Africa region is influenced by monsoon phenomenon which is mainly driven by the surface pressure contrast between the relatively cold waters of the tropical Atlantic Ocean and the Saharan heat low (Knippertz et al.,

2017). This seasonal circulation characterized the wet (summer) and dry (winter) periods in the region. During the dry season (November to February) most of the region is dominated by dry north-easterly winds from the Sahara and the precipitation is confined to the coast, where the sea-breeze circulation provides moister air and produces near-surface convergence. The monsoon starts its development and south-westerly moist winds begin to enter deeper into





the continent producing more clouds and precipitation between July and August, the strong pressure and temperature gradients between the Atlantic Ocean and the Sahara, drive the strong monsoon flow together with south-westerlies reaching higher latitudes, up to 20° N (Knippertz et al., 2015).

Meteorological data from Abidjan, Côte d'Ivoire, are reported in Figure 2. Weekly accumulated precipitation and weekly air temperature means were analysed during the year of field campaigns. Meteorological conditions in Abidjan are also characterized by the monsoon phenomenon which establishes two well defined seasons: a wet season

between March to August and a dry season from November to February. The weekly mean air temperature observed was between 24.6 and 29.4 °C, reaching a maximum during the beginning of the wet season (Figure 2). The precipitation pattern shows an increased rate during the monsoon period, however, this year negative anomalies were observed compared with previous years (Knippertz et al., 2017). Wind patterns observed during the field campaign period showed a predominant contribution from the south-westerly with maximum speed during daytime reaching

up to 13 m.s$^{-1}$. The high wind speed records reported in Abidjan are more significant than those observed in other polluted urban atmospheres (Dominutti et al., 2016; Salameh et al., 2016a; Zhang et al., 2014). Deroubaix et al. (2018) analysed the dispersion of urban plumes from SWA coastal cities. In this study, the inland transport of anthropogenic coastal pollutants together with biomass burning emission was observed and their later northward transport into continental SWA. The proximity of Abidjan to the ocean and the intrusion of the sea-breeze circulation

can facilitate the dispersion processes and, consequently, the urban emissions dilution.

*3.2 VOCs in Abidjan atmosphere*

This analysis relies on the fifteen VOC species already described in Keita et al (2018) and both measured in ambient air and at emission sources. These VOCs include 8 aromatics, 3 monoterpenes, 4 alkanes and isoprene which span a

wide range of reactivity and represent the various types of VOC expected to be released by fossil /non-fossil fuel combustion and biogenic emissions.

3.2.1. Ambient concentrations and spatial distribution

The ambient concentration sum of the fifteen quantified VOCs ranged from 6.25 to 72.13 µg.m$^{-3}$ (see size-coded pie

chart, Figure 3). Higher VOC concentrations were reported in KSI, BIN CRE and PL sites (Figure 3). Details on sampling points can be found in Table 2. The major VOCs are toluene, m+p-xylene, benzene, iso-octane, benzene and limonene. In the overall, anthropogenic VOCs dominated the ambient profiles by a factor of 5 to 20 compared to biogenic ones. BTEX (benzene, toluene, ethylbenzene, and *m+p* and o-xylenes), a subgroup of aromatics VOC, usually makes a significant fraction of VOC burden in urban atmosphere (Borbon et al., 2018; Boynard et al., 2014;

Dominutti et al., 2016). They are emitted by fossil fuel combustion from transport and residential sources and evaporation processes such as fuel storage and solvent uses (Borbon et al., 2018). Here, their contribution ranged from 35% to 76 % of the total VOC burden measured at the ambient sites (Figure 3). Therefore, the following discussion will only focus on BTEX as representative of all measured anthropogenic VOC patterns. Figure 3 reports the total VOC spatial distribution at each site, detailed by the BTEX composition. Firstly, a clear positive eastward

gradient in the total measured VOC concentration is depicted in the Abidjan district. Such spatial variability has been already pointed out by recent studies performed in Abidjan for other atmospheric pollutants (Bahino et al., 2018; Djossou et al., 2018). Not only a similar spatial gradient was observed in ozone concentrations (Bahino et al., 2018)



but also maximum aerosols concentrations were reported near domestic fires (KSI, Figure 1) and landfill sites (AD, Figure 1), indicating that open-air burning activity could be a significant contributor to air pollution, by a factor of 3

and 5, respectively. The commonalities in spatial distribution seem to be also related to the well-established south-western wind direction (Figure 2a). This wind pattern was already described in the West Africa region when the urban plumes of the main coastal cities were analysed. The dispersion and transport of the urban plumes from Abidjan were predominantly directed into north- to eastward directions, reaching a distance up to 200km from the city throughout the 24 hours model simulations (Knippertz et al., 2017).

Second, when looking at BTEX, m+p-xylene and toluene dominate the ambient profiles of BTEX. However, their contributions, as well as these from other BTEX, varied according to the sampling sites. The heterogeneity in their spatial distribution could be related to the main activities that are involved in the emission of these compounds since the provided dilution rate is assumed to be constant. It can be noted a similar BTEX distribution profile between some specific sites, such as AT and BIN and KSI, PL and CRE, for instance.

The mean ambient concentrations observed in Abidjan were compared with those performed in other cities worldwide (Figure 4). Globally, a good agreement is observed with the mean VOC concentrations showing the same distribution in most of the cities, except for Karachi and to a lesser extent São Paulo, where more substantial VOC levels were reported (Barletta et al., 2005; Dominutti et al., 2016). Our results depict that ambient VOC levels in Abidjan are noticeably similar when compared with northern mid-latitude megacities., suggesting that emissions from fossil fuel

combustion dominate over other specific sources.

### 3.2.2. Ambient composition vs. emission source profiles

A comparative approach was carried out between ambient and source measurement composition with the purpose to detect emission source fingerprint in ambient VOC profiles. Figure 5 reports the relative mass contribution of VOC

profiles observed at the nine urban sites together with those obtained from the emission sources. Toluene, xylenes, and 124-TMB presented the highest variability in concentration among anthropogenic VOCs and limonene among biogenic VOC at emissions and ambient sites. Trimethylbenzenes (124-TMB, 135-TMB, and 123-TMB), mainly observed in road transport emissions, display a dissimilar profile showing higher fractions from sources than those observed at ambient sites (Figure 5). These differences might be related to the short lifetime of these compounds

(around 4 hours), with a reaction rate within 1.8 to 8.8 ($x10^{-15}$ $cm^3$ $molecule^{-1}$ $s^{-1}$, (Atkinson and Arey, 2003a)). Their reactivity implies a faster reaction in the atmosphere and the losses of these species from emission to the receptor. ambient profiles showed in most of the cases, higher contributions of alkanes, terpenes, and isoprene. It is noteworthy the presence of terpenes and isoprene in the profile of all the emission sources, mainly in those associated to domestic burning processes, such as charcoal burning, waste burning and fuelwood burning (Figure 5). The terpene emissions

from biomass burning were already pointed out in several studies, as common species emitted by the combustion processes (Gilman et al., 2015; Simpson et al., 2011). Additional studies performed on African biomass emissions also reported concentrations of limonene and α-pinene higher than isoprene (Jaars et al., 2016; Saxton et al., 2007). This pattern is similarly observed in our study where considerable limonene and pinenes concentrations were measured in ambient sites.

For the selected VOC species, aromatic compounds represent the higher fraction from ambient and source profiles, contributing from 31 to 75 per cent (Figure 5). Comparing the same VOC species in emissions versus ambient

profiles, we found a similarity with the two-wheelers and domestic fires profiles like FW and CH sources. Notwithstanding that, the VOC ambient profiles obtained from the nine sites did not show a contrasted difference despite the dissimilarities in the activities conducted nearby.


*3.3. VOC molar mass contribution from Abidjan sources.*

Emission ratios (ER) were computed for each VOC identified related to CO concentrations measured at the emission (Table SM1). VOC to CO ERs were computed to facilitate the integration of this new dataset into models and emission studies. To evaluate the composition and intensity of SWA emissions we analysed here their contribution

regarding molar mass related to VOC emission ratios. For this analysis, an extensive VOC database of 56 species was considered, including 12 terpenes, VOCs of intermediate volatility (IVOCs from C11–C16 n-alkanes), ketones and carbonyl compounds for all sources (Table 1 and Table SM1). Species groups were classified in agreement with GEIA groups (Huang et al., 2017) respecting the chemical function of each VOC family (Table SM2). In this way, ER were also grouped by VOC family from individual values obtained (Table SM1). Figure 6 shows the relative

molar mass profiles of each source measured in Abidjan. As already depicted in previous section, aromatic (VOC13-VOC17) molar masses ruled the distribution of most of the sources, which jointly range from 26 % to 98 % of the total ER molar mass. The control of these compounds is predominantly observed in gasoline-fuelled vehicles, like LDGV and TW sources and diesel light-duty vehicles (LDDV). Alkanes (VOC5+VOC6) also comprise noticeable molar mass fraction, dominating in TW2T, HDDV and charcoal related sources (by a 40, 47 and 53% respectively).

Since the VOCs of intermediate volatility (IVOCs) do not have a specific classification, they were integrated in the group of heavy alkanes (VOC6). A considerable IVOCs contribution from the emission of HDDV sources was observed, where IVOCs dominate the VOC6 fraction by a 30% (considering that VOC6 represents the 47% of the total emissions from this source).

Interestingly, terpenes (VOC11) reported an 11%, 13% and 22% contribution in FW, HDDV and WB sources,

respectively (Figure 7b-c). The presence of terpenes in biomass burning sources was already pointed out as the most important compounds together with furans and aromatics, in chamber experiments (Koss et al., 2018). Nevertheless, to the extent of our knowledge, their presence in road transport or open waste burning in SWA emissions remains unexplored. Regarding OVOCs (VOC22), in general, they are emitted in a smaller fraction (less than 7%) apart from HDDV which provide 11% of the total molar mass. Previous studies have been reported the OVOCs as the main

contribution fraction in biomass burning emissions (Akagi et al., 2011; Gilman et al., 2015; Yokelson et al., 2013). A different pattern is observed in this study, likely related to the limitation of VOCs species measurements by the analytical method deployed, which allows the quantification of a limited number aldehydes (>C6). Sekimoto and co-workers also analysed the VOC emission profiles depending on the pyrolysis temperature, showing enrichment of terpenes and non-aromatic oxygenates under high-temperature conditions and an increase in oxygenated aromatics

under low-temperature fires (Sekimoto et al., 2018).

Four sources, representing the leading sectors in the region (road transportation, waste burning, and charcoal emissions, were selected (TW2T, HDDV, WB, and CH) to analyse absolute contribution and potential impacts related to African emissions. Figure 7 (a-d) shows the relative composition and the total molar mass of the measured VOC ($\mu g\ m^{-3}$) emitted per $ppm_v$ CO. TW2T sources disclosed the highest molar mass emissions ($4680 \pm 512\ \mu g\ m^{-3}\ ppm_v$

$CO^{-1}$, Figure 7a-d), 10 to 200 times higher than any other source here analysed.





While aromatics(VOC13-VOC17) seem to dominate the molar mass fraction of ERs for most of the sources, their contributions are dissimilar, governed by benzene (VOC13) and toluene (VOC14) in burning-related sources and by xylenes (VOC15) and trimethylbenzenes (VOC16) in traffic-related ones. Despite the non-controlled sampling and the restricted number of VOC species attained by our study, they represent a meaningful dataset for the first

estimation of ER in SWA region.

*3.4 Implications on atmospheric reactivity*

3.4.1 VOC-OH reactivity

The estimation was based on the calculated OH reactivity by using the reaction rate coefficient (Atkinson and Arey,

2003b; Manion et al., 2015) of each VOC species, as detailed in Equation 3, and then aggregated into VOC families. Estimations for all the sources were performed; however, only the results for the selected sources (TW2T, HDDV, WB, and CH) are discussed here. Figure 7(e-h) shows the fractional contributions and total VOC-OH reactivity per $ppm_v$ of CO for the selected sources. The largely highest total reactivity is observed from the emissions of TW2T ($488 \pm 43$ $s^{-1}$ $ppm_v$ $CO^{-1}$), outpacing other sources within a factor of 7 to 170. This disclosed difference is related to

the high ER observed for the more reactive species, like terpenes (VOC11) and C8- and C9-aromatics (VOC15 and VOC16 respectively). Terpenes (VOC11) and aromatics (VOC13-VOC17) altogether are the dominant sink of OH, contributing with 47 to 87% of the total calculated OH reactivity. Individually, terpenes ruled the OH-reactivity in open waste burning emissions (76%) and heavy-duty diesel vehicles (60%) (Figure 7f-g). In charcoal burning emissions it may be noted the fractional contribution of aldehydes (VOC22, 13%) and higher alkanes (VOC6, 28%).

The modest presence of alkenes in the VOC-OH fractional analysis, well-known for their high reactivity effects, is related to the limitation of the sampling method which restricts the collection of light alkenes species. We might expect a high contribution from alkenes adding to the terpene burden.

3.4.2 Ozone formation potential

The estimation of ozone formation was evaluated by using the POCP scale, developed by Derwent (Derwent et al., 2010a) as expressed in equation 4. Overall, the fractional ozone formation distribution is dominated by aromatics (VOC13 to VOC17) in all the sources, by 38 to 63 per cent. Alkanes (VOC6) represent an essential contribution in charcoal burning, HDDV, and TW2T, accounting for 45, 28 and 26 per cent, respectively. It is important to note the terpenes (VOC11) contribution, not only from burning sources but also from the road transportation sector (Figure

7i-l). Aldehydes (VOC22) are well-known due to their high reactivity in the atmosphere (Atkinson and Arey, 2003a; Sommariva et al., 2011), and some of these species have been shown to have a large impact on ozone formation and chemistry. In our estimation, we can observe the contribution of these compounds mainly from diesel (HDDV) and charcoal burning sources (CH).

The total potential ozone was computed for each source, showing a mostly dominant contribution from TW-2T

(80343 POCP $ppm_v$ $CO^{-1}$), 13, 24 and 150 times higher than the potential impact in ozone formation derivate from the HDDV, WB and CH emissions, respectively.

3.4.3 SOA formation potential



Figure 7 (m-p) shows the composition and mean SOA formation potentials of VOC families emitted by each selected source. As it can be expected charcoal burning reports the lowest SOAP (335 SOAP per $ppm_v$ $CO^{-1}$), compared to TW2T, HDDV and WB sources whose SOAPs values are 147, 10 and 9 times greater, respectively. It may be noted that, globally, aromatics (VOC13-VOC17) governed the SOA formation in our estimations, by 98 to 72 per cent. Interestingly, terpenes (VOC11) represented a minor contribution in the SOA formation, presenting SOAP index lower than those for aromatics species, representing approximately 20% of the SOAP for toluene (VOC14). Despite the well-known role of biogenic VOCs as SOA precursors (Ait-Helal et al., 2014), the method used here is not able to correctly quantify their contributions in SOA formation. The differences between SOAP values and measured aerosols yield were already pointed out by Gilman and co-authors, who performed some sensitivity tests in order to bring in line SOAP and aerosols yields (Gilman et al., 2015). We performed here the same analysis, by adjusting the SOAPs terpenes values 10% higher. The results in total SOAP per $ppm_v$ of CO, did not show considerable increases in any of the sources, expanding the total SOAP up to 1%. Similar results were observed for fractional distribution, where the changes in terpenes SOAPs (VOC11) did not show any substantial change in the VOC contribution for SOA formation. These findings are in agreement with those identified in the study of Gilman et al. (2015), suggesting an underestimation in the fractional contribution of terpenes in the potential formation of organic aerosols over SWA region.

### 3.5 Quantification of VOC emissions

The VOC emissions were estimated from our measurements, considering the extensive dataset of fifty-six compounds. For that, VOC emissions were estimated by bringing the emission factors obtained in near-source measurements together with the statistical IEA activity data (International Energy Agency (IEA)) available for the different sources (Keita et al., 2018). To compute the emission factors the equation 1 was used, considering all the VOC species measured and including the mass fraction of each fuel (fc) obtained from the literature. Additionally, the differences in motorizations and the fleet age have been considered, as well as the fleet distribution by calculating the equivalent vehicular fleet as described in Keita et al. (2018). In the residential profile we integrated those emissions measured from CH, CHM and FW sources, commonly observed at residential sites in Abidjan. Afterwards, the mean road transportation and residential profiles for Cote d'Ivoire were computed and compared with two referenced global inventories, EDGAR v4.3.2 and MACCity (Granier et al., 2011; Huang et al., 2017).

The first insight on these estimations was provided in the work of Keita and co-authors, by analysing discrepancies and commonalities with the emissions estimated by the EDGAR v4.3.2 global inventory (http://edgar.jrc.ec.europa.eu) In this previous study, the selected VOC database disclosed a considerable disagreement for the road transport sector by a factor of 50 (Keita et al., 2018), only considering fifteen VOC species. Here, we report the mean road transport profile by considering the complete VOC dataset including the 56 compounds analysed, incorporating aldehydes, IVOCs and terpenes species. Mean residential emissions are also integrated and compared with those from Edgar v4.3.2 inventory. Additionally, we incorporate the residential and road transport profiles provided by MACCity inventory (Granier et al., 2011) available in the ECCAD-GEIA database (http://eccad.aeris-data.fr). The main differences between both global inventories are related to the speciation level for VOCs families. MACCity considered all the aromatics in the same VOC group; thus, we provide here the sum of VOC13 to VOC17 families (Table SM2) to compare it with the aromatics group from this inventory.



Figure 8 exhibits the speciated emissions obtained for Côte d'Ivoire along with those provided by emission inventories. Globally, the discrepancies already highlighted in the previous analysis are exacerbated by introducing the complete VOC database. Residential profiles showed a disparity in total VOC emissions by a factor of 14 with Edgar v4.3.2 and by a factor of 43 compared with MACCity (Figure 8a). In terms of composition, the main differences observed in residential profiles are related to VOC22 group (aldehydes), which discloses a more significant contribution in the EDGAR inventory, accounting for 64% of the total emission, 5 times higher than our estimation. It should be noted that there is also a disparity in the contribution from aromatics (sum of VOC13 to VOC17) and alkenes (VOC12) which reveals a more substantial influence in the MACCity profile (58 and 22 per cent, respectively) (Figure 8a). This disparity could be related to the few VOC species that were analysed for the VOC12 group in our study. Nevertheless, aromatics dominate the fractional contribution on the updated emission inventory (39%), especially by toluene (VOC14) and C8-aromatics (VOC15) (11 and 10 %, respectively). Alkanes (>VOC6 alkanes) disclose a more significant contribution in the residential profile, in which IVOCs contribute by 20% of the total alkanes estimated.

Regarding road transportation sector, the comparison with global inventories depicted a factor 100 and 160 times lower in total emissions for Edgar and MACCity profiles, respectively (Figure 8a). A somewhat agreement is observed with speciation (Figure 8b). Aromatics and alkanes govern maximal contributions from all profiles in different proportions. Our estimations report the more significant contributions in C8-aromatics (VOC15), C9-aromatics (VOC16) and toluene (VOC14), ranging for 25, 14 and 10 per cent, respectively (Figure 8c and Figure 9). Whereas Edgar v4.3.2 discloses a contribution of 9% for VOC15, 3.5 % for VOC16 and 13% for VOC14 (Figure 9). Road transport profiles also reproduce the anomalies in the VOC12 contribution observed in the residential sector, presenting greater emissions in the global inventories. The comparison between both inventories also depicted considerable discrepancies, rising a factor of 3.

A similar profile is observed for higher alkanes (VOC6) which present a quite analogous contribution between our estimation and Edgar emissions (34 and 37 per cent, respectively; Figure 8b). Nevertheless, the alkanes (VOC5+VOC6) contribution in MACCity profiles prevails on road transport emissions accounting for 62%.

Interestingly, terpenes and isoprene emissions can be denoted in both sectors in Côte d'Ivoire inventory (VOC11 and VOC10). Despite the reduced contribution of these species (9 and 4 per cent in residential and road transport), the underestimation of them in the emission from anthropogenic sources could have consequences in the atmospheric chemistry. Since the reactivity is specific for each VOC, the inaccuracies in the speciation also have implications on the estimation of their impacts. Specifically, for terpenes (VOC11), it can be noted the contribution in the kOH reactivity, accounting for 42% in the residential sector and 28% in road transport reactivity (Figure 8c). Even though the total OH reactivity in all profiles are rather similar, the alkenes fraction in this study is not well-represented that could increase the contribution in terms of reactivity.

Figure 9 also displays the residential and road transportation profiles obtained from Côte d'Ivoire compared with Edgar v4.3.2 profiles for Europe. Noticeably in our estimations, road transport and residential sectors displayed comparable total emissions, whereas those from Edgar inventory depicted a dissimilarity by a factor of 8 (86.1 vs 12.1). Dissimilar tendencies are also observed when comparing Edgar total emissions for Europe and Cote d'Ivoire, where the former present the larger emissions (198 vs 86 and 433 vs 12.1) We highlight here the substantial



differences in total emissions, outpacing those estimated for Europe by a factor of 3 for road transport and by a factor of 6 for residential sector (433 and 198 Gg year-1, respectively).

The dearth of measurements and source profile data in Africa was previously pointed out in the development of the EDGAR inventory, which leads to considering the priority of this region for future inventory improvements (Huang

et al., 2017). Our results emphasize the first insights obtained in the work of Keita et al. (2018) where the considerable underestimation of anthropogenic VOCs emissions by emission inventories was revealed. Even though our VOC database is not extensive for all the species emitted by the sources analysed, the incorporation of new VOC species reinforces the usefulness of in situ measurements under real conditions to derive realistic emission factors and subsequent estimation of representative emission profiles.

*3.6 Anthropogenic emissions of terpenes, IVOCs and aldehydes in SWA*

As it was previously highlighted, the presence of terpenes commonly emitted by biogenic sources was observed in the emissions of anthropogenic sources. Global emission inventories wholly neglect these emissions; however, they could have considerable effects in the atmospheric chemical processing, by producing secondary pollutants in the

555 atmosphere. Figure 10a reports the fractional distribution of terpenes in the several emission sources analysed. Main contributions are associated to the emissions from waste burning (WB, 47%), two-wheelers (TW2T, 20%), wood burning (FW, 17%) and charcoal making (CHM, 14%) sources. The total annual emission estimated for these compounds, which represents 334 Gg. year$^{-1}$ and 11% of the total emissions, cannot be neglected. Evaluating the distribution by terpenes species among the emission sources, a different pattern can be noted (Figure 11). While

terpenes emissions from road transport are mainly dominated by α-ocimene and α-terpinolene, limonene and isoprene are controlled by wood-burning sources. The main wood burning species in Côte d'Ivoire are Hevea (*Hevea brasiliensis)* and Iroko (*Milicia excelsa*), which are widely used in urban domestic fires for cooking, heating and other services (Keita et al., 2018). In our study, we only present the results obtained from Hevea, a tropical African hardwood, characterised as a monoterpenes emitter species (Bracho-Nunez et al., 2013; Wang et al., 2007). The

principal monoterpenes compound naturally emitted by hevea species are sabinene, limonene, and α-pinene (Bracho-Nunez et al., 2013). The isoprene emissions from non-isoprene emitting species were already observed in biomass burning studies, which indicates that isoprene is formed during the combustion process (Hatch et al., 2015).

As it can be noted in Figure 11, isoprene emissions are also impacted by vehicles, mainly TW sources, and camphene and β-pinene emissions by HDDV sources. The anthropogenic sources of isoprene have been documented in urban

areas, mainly associated with traffic emissions (Borbon et al., 2001; von Schneidemesser et al., 2011). However, to the best of our knowledge, no previous studies had never analysed the presence of monoterpenes from road transportation sources. A-pinene and β-pinene emissions are ruled by charcoal burning fires, which also contribute in some fraction to the emissions of isoprene and limonene. In contrast, charcoal making emissions are dominated by γ-terpinene and isoprene. The results from biomass burning sources provided here were obtained from non-

controlled experiments, which did not allow to evaluate the divergences between the emissions from each combustion phase (pyrolysis, flaming and smouldering). Further investigation is needed for a better understanding of these differences and the characterization of all the combustion phases.

VOCs of intermediate volatility are suspected to be efficient precursors of secondary organic aerosol (SOA) (Seinfeld and Pandis, 2006 and references therein). However, as it was discussed in the section 3.4.3, our method was not able



to resolve the differences between VOC families and most of the SOAP was assigned to aromatic compounds (up to 98%). Figure 11b reports the fractional contribution and total emissions of IVOCs. CHM, FW, HDDV, and TW represent the primary sources of these compounds, accounting for 58, 15, 12 and 11 per cent of the total, respectively. Despite their lower emissions compared to aromatics or terpenes, IVOCs are estimated to account for 80Gg/year emissions in Côte d'Ivoire. A recent study observed that fine particles in Abidjan are three times higher than the concentrations recommended by the World Health Organization (Djossou et al., 2018). Hence, a better understanding of the aerosol precursors and formation processes is substantial for the later reduction of their concentrations in the urban atmosphere.

Oxygenated compounds were previously indicated as essential species in the emissions from burning sources (Gilman et al., 2015; Hatch et al., 2015; Koss et al., 2018; Wiedinmyer et al., 2014). In addition, oxygenated compounds like non-aromatics oxygenated were the dominant fraction in burning emission sources including a range of functional groups, of which alcohols and carbonyls were the most abundant (Koss et al., 2018; Stockwell et al., 2015). Figure 11c shows that aldehydes emissions are mainly governed by charcoal fabrication (CHM), two-wheelers (TW) and wood burning sources (Figure 11c). In our study, the quantified aldehydes represent only 5.5% of the total emissions of the country (170 Gg/year). However, they can be essential compounds concerning reactivity and ozone formation. Hence, further analysis of oxygenated compounds together with furans and other nitro-oxygenated compounds need to be addressed in future campaigns, to improve not only the quantification of these compounds but also a better identification of the African tracers from biomass burning processes.

## 4. Summary and conclusions

The present study reports for the first time a chemically detailed range of VOCs including C5-C16 alkanes, monoterpenes, alkenes, aromatics and carbonyls compounds by using sorbent tubes during an intensive field campaign in Abidjan, SWA. We present here an original dataset integrating main emission sources and ambient measurements from nine representative sites and covering the urban spatial distribution of VOCs in Abidjan. The overview of urban air composition resulted in a VOC profile which is mostly affected by biomass burning and traffic emissions. The highest concentrations were observed near domestic fires, landfill fires and traffic sites, in agreement with the results reported in previous studies, where gas-phase and aerosols pollutants were measured (Bahino et al., 2018; Djossou et al., 2018).

The establishment of emission ratios is an important metrics to constraint the estimations provided by global emission inventories. Emission ratios from regional-specific emission sources were established here and later used for the analysis of fractional molar mass contribution and the estimation of potential VOC-OH reactivity, ozone and secondary organic aerosol formation. The distribution of VOC emissions (magnitude and composition) was different for each source evaluated. Two wheelers and heavy-duty sources presented the most significant total molar mass emissions while charcoal-burning the lowest. The sources related to burning processes, such as waste and wood burning, also presented significant contribution in VOCs emissions. These sources represent common activities developed in Abidjan, suspected to contribute with a large quantity of VOC emissions over the SWA region.

Regarding VOC speciation, molar mass contributions were mostly dominated by aromatics and alkanes compounds. Since a few alkenes species were identified, aromatic ruled the ozone formation potential and the SOA formation potential. However, the SOA metrics applied here was not able to accurately analyse the contribution from other




important SOA precursors, such as monoterpenes. Notwithstanding that, monoterpenes can have a significant

contribution in the VOC-OH reactivity from some sources like WB and a higher contribution from the whole alkanes

species could be expected in the total reactivity.

In order to estimate the magnitude of VOC emissions in Côte d'Ivoire, emissions factors were determined from in-situ VOC database. Road transportation and residential profiles were obtained and compared with those reported in global emission inventories (MACCity and Edgar). Our results revealed a discrepancy up to a factor of 43 and 160

for residential and transport profiles when compared with both referenced inventories. The high levels of VOC emissions obtained for Côte d'Ivoire outpace European emissions up to a factor of 6. Interestingly, monoterpene emissions were observed in anthropogenic sources emissions from biomass burning to road transportation sources contributing with up to 340 Gg/year in the annual emissions. These compounds, are generally missing in the global anthropogenic emission profiles which would imply an underestimation of their air quality impacts. This

underestimation is not only expected for Côte d'Ivoire but for all West Africa countries.

This study in the framework of the DACCIWA project, allowed us for the first time to identify and quantify several VOCs in ambient air and at emission in Abidjan, Côte d'Ivoire. Our results provide significant constraints for the development of more realistic regional emission inventories. A continuous effort is needed to collect new ambient and emissions data in West Africa countries for all critical atmospheric pollutants.


**Acknowledgments**

This work has received funding from the European Union Seventh Framework Programme (FP7/2007-2013) under grant agreement number 603502 (EU project DACCIWA: Dynamics-aerosol-chemistry-cloud interactions in West Africa). P Dominutti acknowledges the Postdoctoral Fellowship support from the Université Clermont Auvergne and

thanks the grant received from the CAPES program from the Ministry of Education of Brazil, during 2015-2016. Thierry Leonardis is thanked for the contribution in the analysis of VOC sorbent tubes, graciously performed at the SAGE Department at IMT Lille Douai (France).

*Data availability*. After the DACCIWA embargo period, the data used in this study will be available on the SEDOO

database

*Competing interests*. The authors declare that they have no conflict of interest.





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



**List of figures**

**Figure 1**. Geographical location of Abidjan, Côte d'Ivoire and spatial distribution of ambient VOC measurements. Red stars indicate the VOC measurement sites and the blue squared point represents the meteorology site. More information about the ambient site is detailed in Table 2.

**Figure 2**. Meteorological data observed from Abidjan, Côte d'Ivoire. The figure represents a) the weekly accumulated precipitation and weekly mean air temperatures and b) the wind speed and direction observed, during field campaigns. Data was downloaded from the National Centers for environmental information site (NCDC), NOAA and recorded at Abidjan International Airport (see location in Figure1).

**Figure 3.** Spatial distribution of VOC measured in ambient sites in Abidjan size-coded by the sum of VOCs (in $\mu g/m^3$) and color-coded by the relative contribution of BTEX compounds, namely benzene (Benz), toluene (Tol), ethylbenzene (EthylB), $m+p$-xylene ($m+p$-xyl) and o-xylene (o-xyl). Values showed in each pie-chart represent the total VOC concentration measured at the sampling point. Ambient site names and characteristics are presented in Table2.

**Figure 4**. Boxplot showing the VOC concentrations ($\mu g.m^{-3}$) in Abidjan ambient sites (upper panel). The middle line in each box plot indicates the median (50th percentile), the lower and upper box limits represent the 25th and 75th quartiles and the whiskers the 99% coverage if the data have a normal distribution. The low panel shows the mean concentrations reported in other cities worldwide, such as Abidjan (this study), Paris (AIRPARIF, 2013), São Paulo (Dominutti et al., 2016), Beirut (Salameh et al., 2014), Karachi (Barletta et al., 2002) and South Africa (Jaars et al., 2014).

**Figure 5**. Relative concentration comparison between ambient measurements and emission sources profiles of VOCs measured in Abidjan, Côte d'Ivoire. Orange-yellow based-colours represent alkanes, blue based-colours aromatics, and green-based colours terpenes and isoprene contributions.

**Figure 6.** Distribution of relative emission ratios in molar mass basis from the emission sources analysed in Abidjan, aggregated in VOC families (table SM2). The emission sources under analysis are heavy-duty diesel vehicles (HDDV), two-wheelers two strokes (TW-2T), two-wheelers four strokes (TW-2T), light-duty diesel vehicles (LDDV), Light-duty gasoline vehicles (LDGV), charcoal burning (CH), fuelwood burning (FW), charcoal making (CHM) and landfill waste burning (WB).

**Figure 7**. VOC emission ratios contributions to (a)–(d) the measured molar mass, (e)–(h) OH reactivity, (i)– (l) relative ozone formation potential POCP and (m)-(p) relative SOA formation potential, aggregated in VOC families. Absolute totals for each source are shown below each pie chart in the respective units.

**Figure 8**. Comparison of VOC emission profiles for Côte d'Ivoire between the emissions estimated from the measurements of this study (Keita et al., 2018), Edgar v4.3.2 (Huang et al.; 2017) and MACCity inventory (Granier et al., 2011).The profile analysis integrates road transportation and residential sectors based on the sector activity for 2012. a) Absolute emissions in Tg/year, b) relative mass contribution, and c) relative mass reactivity considering 100 Tg of emissions weighted by the $k$OH reaction rate calculated for each VOC family.

**Figure 9**. VOC emission profiles considering all the VOC families calculated from the measurements of our study and compared with the global Edgar v4.3.2 inventory (Huang et al.; 2017). The comparison integrates road transportation (RoadT and RT) and residential (Resid) sectors in Côte d'Ivoire and Europe for the year 2012. Absolute emissions are expressed in Gg/year for each VOC group.

**Figure 10**. Total estimated emissions and relative distributions in the anthropogenic sources measured in Abidjan, Côte d'Ivoire. Each pie chart represents a VOC family, like a) Terpenes, b) IVOCs and c) Aldehydes.

**Figure 11**. Distribution of monoterpenes and isoprene in the emission sources measured in Abidjan. The values represent the percentage of each biogenic VOC over the total emission estimated for these species.





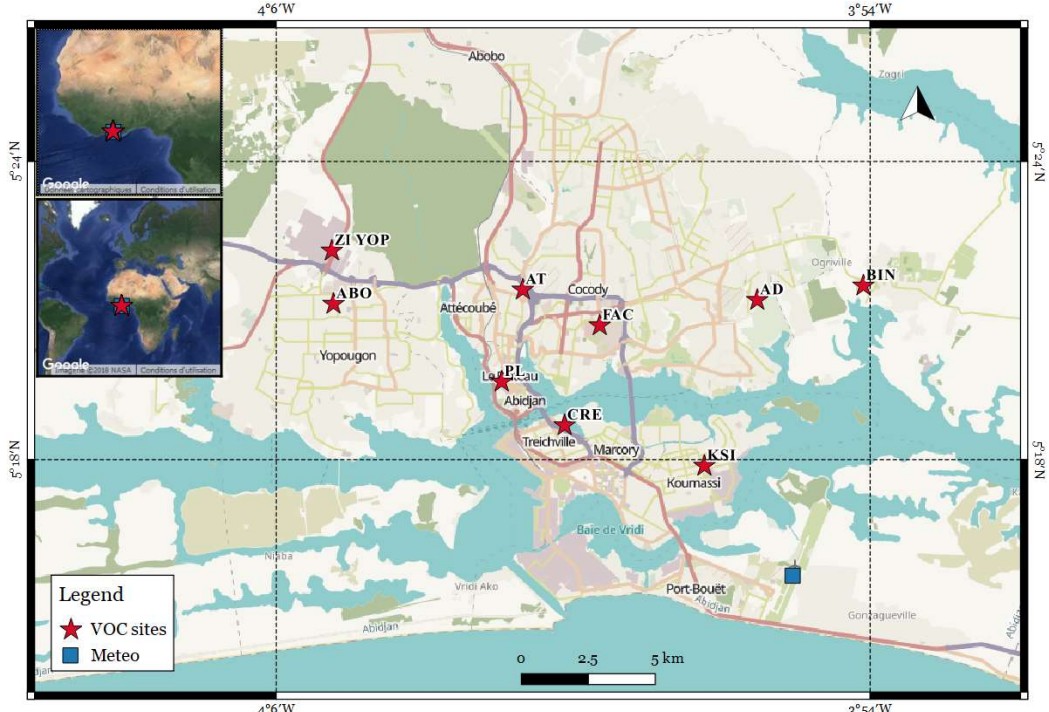

**Figure 1**. Geographical location of Abidjan, Côte d'Ivoire and spatial distribution of ambient VOC measurements.
Red stars indicate the VOC measurement sites and the blue squared point represents the meteorology site. More
information about the ambient site is detailed in Table 2.





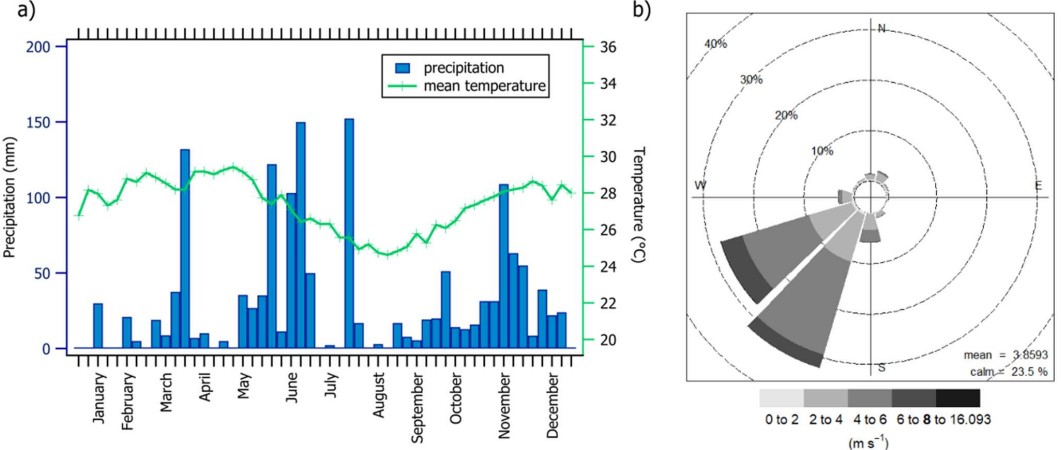

**Figure 2**. Meteorological data observed from Abidjan, Côte d'Ivoire. The figure represents a) the weekly
accumulated precipitation and weekly mean air temperatures and b) the wind speed and direction observed, during
field campaigns. Data was downloaded from the National Centers for environmental information site (NCDC),
NOAA and recorded at Abidjan International Airport (see location in Figure1).






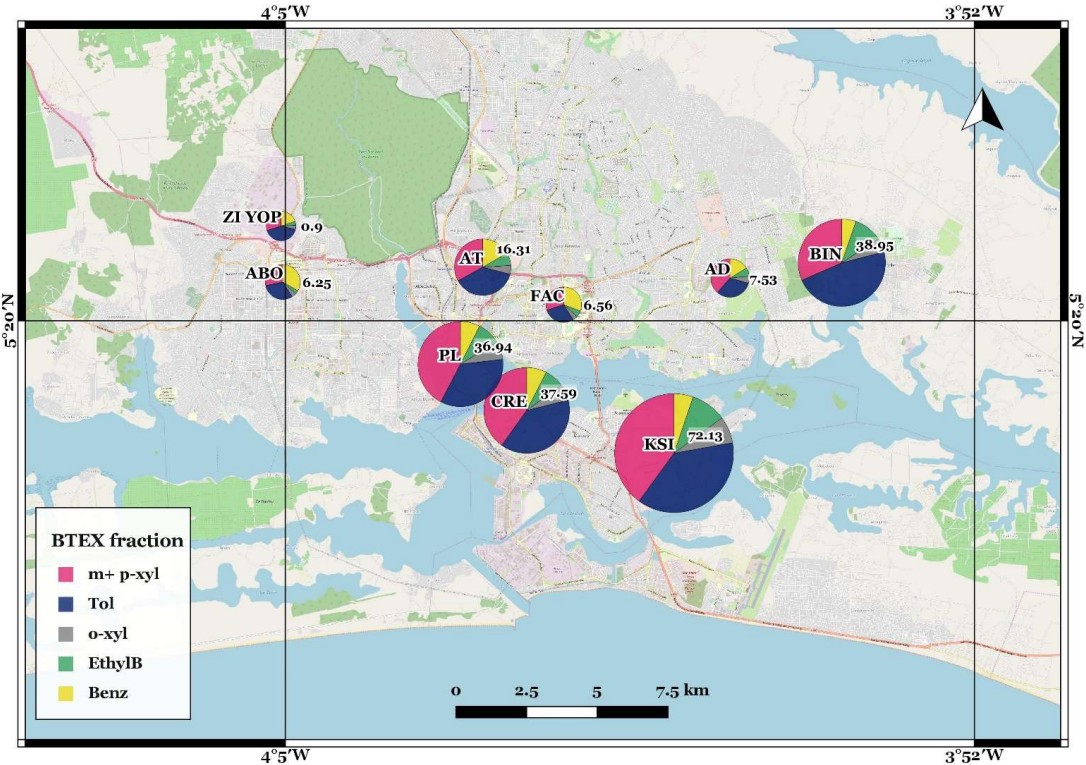

**Figure 3.** Spatial distribution of VOC measured in ambient sites in Abidjan size-coded by the sum of VOCs (in µg/m$^3$) and color-coded by the relative contribution of BTEX compounds, namely benzene (Benz), toluene (Tol), ethylbenzene (EthylB), $m+p$-xylene ($m+p$-xyl) and o-xylene (o-xyl). Values showed in each pie-chart represent

the total VOC concentration measured at the sampling point. Ambient site names and characteristics are presented in Table2.



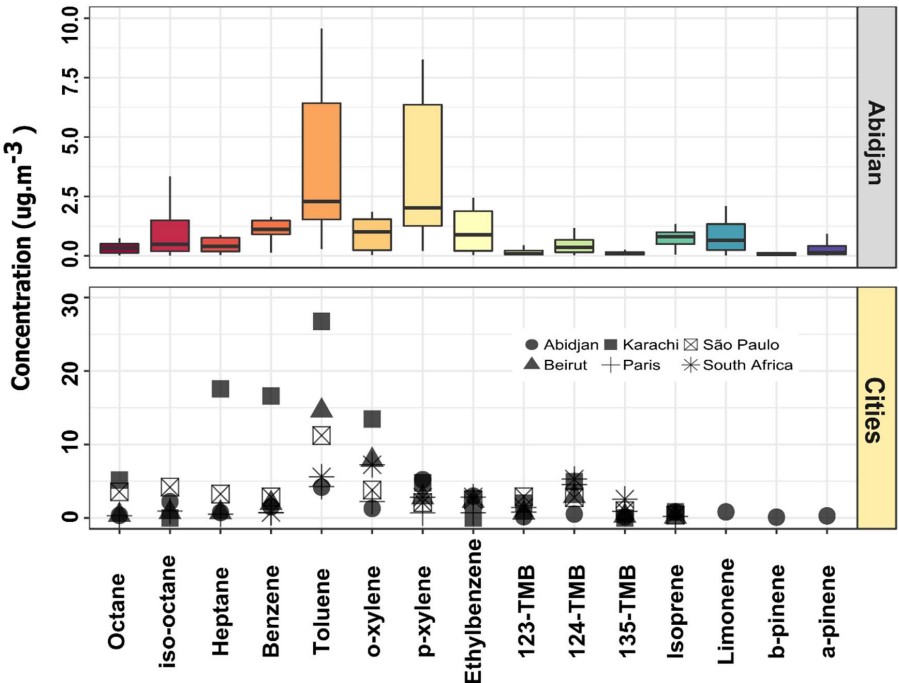

**Figure 4**. Boxplot showing the VOC concentrations (μg.m⁻³) in Abidjan ambient sites (upper panel). The middle line in each box plot indicates the median (50th percentile), the lower and upper box limits represent the 25th and 75th quartiles and the whiskers the 99% coverage if the data have a normal distribution. The low panel shows the mean concentrations reported in other cities worldwide, such as Abidjan (this study), Paris (AIRPARIF, 2013), São Paulo (Dominutti et al., 2016), Beirut (Salameh et al., 2014), Karachi (Barletta et al., 2002) and South Africa (Jaars et al., 2014).





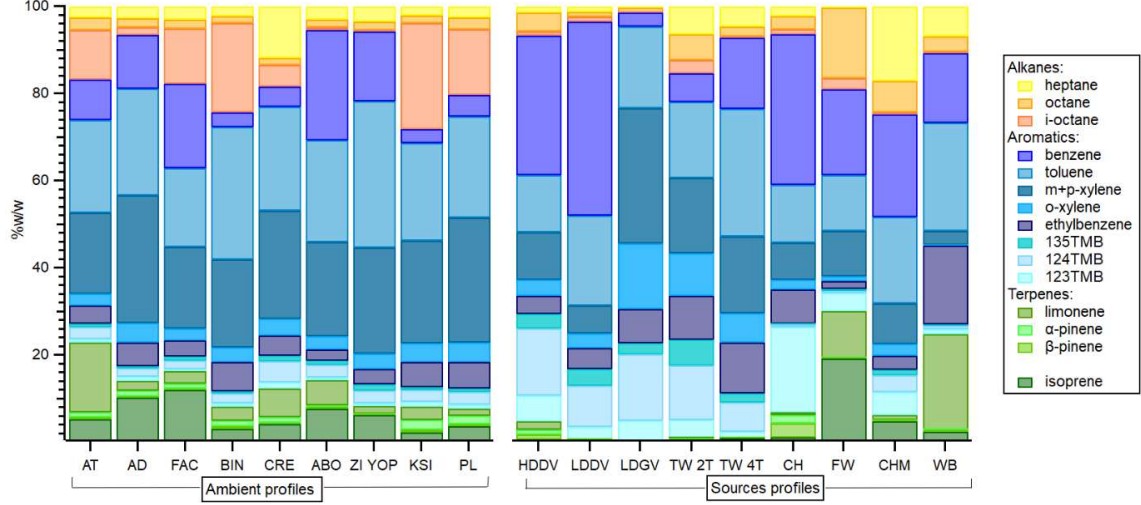

**Figure 5**. Relative concentration comparison between ambient measurements and emission sources profiles of VOCs measured in Abidjan, Côte d'Ivoire. Orange-yellow based-colours represent alkanes, blue based-colours aromatics, and green-based colours terpenes and isoprene contributions.




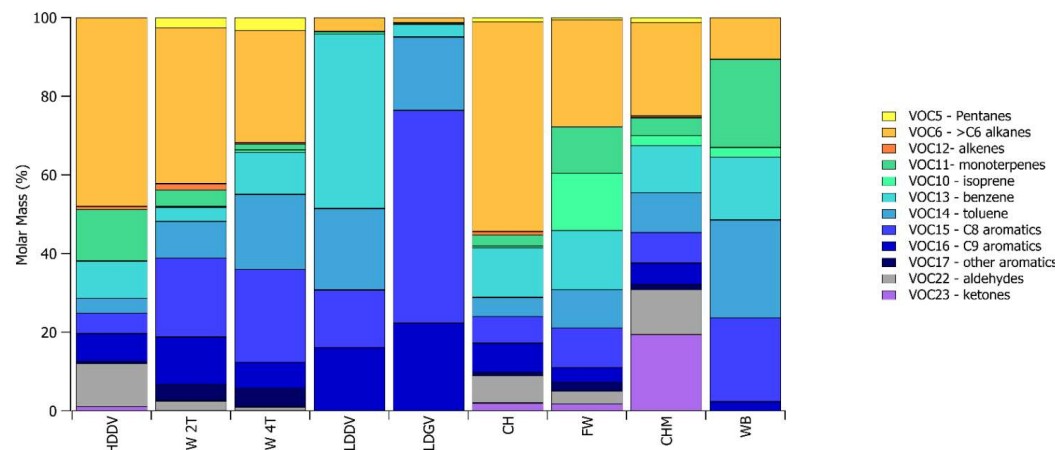

**Figure 6.** Distribution of relative emission ratios in molar mass basis from the emission sources analysed in Abidjan, aggregated in VOC families (table SM2). The emission sources under analysis are heavy-duty diesel vehicles (HDDV), two-wheelers two strokes (TW-2T), two-wheelers four strokes (TW-2T), light-duty diesel vehicles (LDDV), Light-duty gasoline vehicles (LDGV), charcoal burning (CH), fuelwood burning (FW), charcoal making (CHM) and landfill waste burning (WB).




**Figure 7**. VOC emission ratios contributions to (a)–(d) the measured molar mass, (e)–(h) OH reactivity, (i)–(l) relative ozone formation potential POCP and (m)-(p) relative SOA formation potential, aggregated in VOC families. Absolute totals for each source are shown below each pie chart in the respective units.



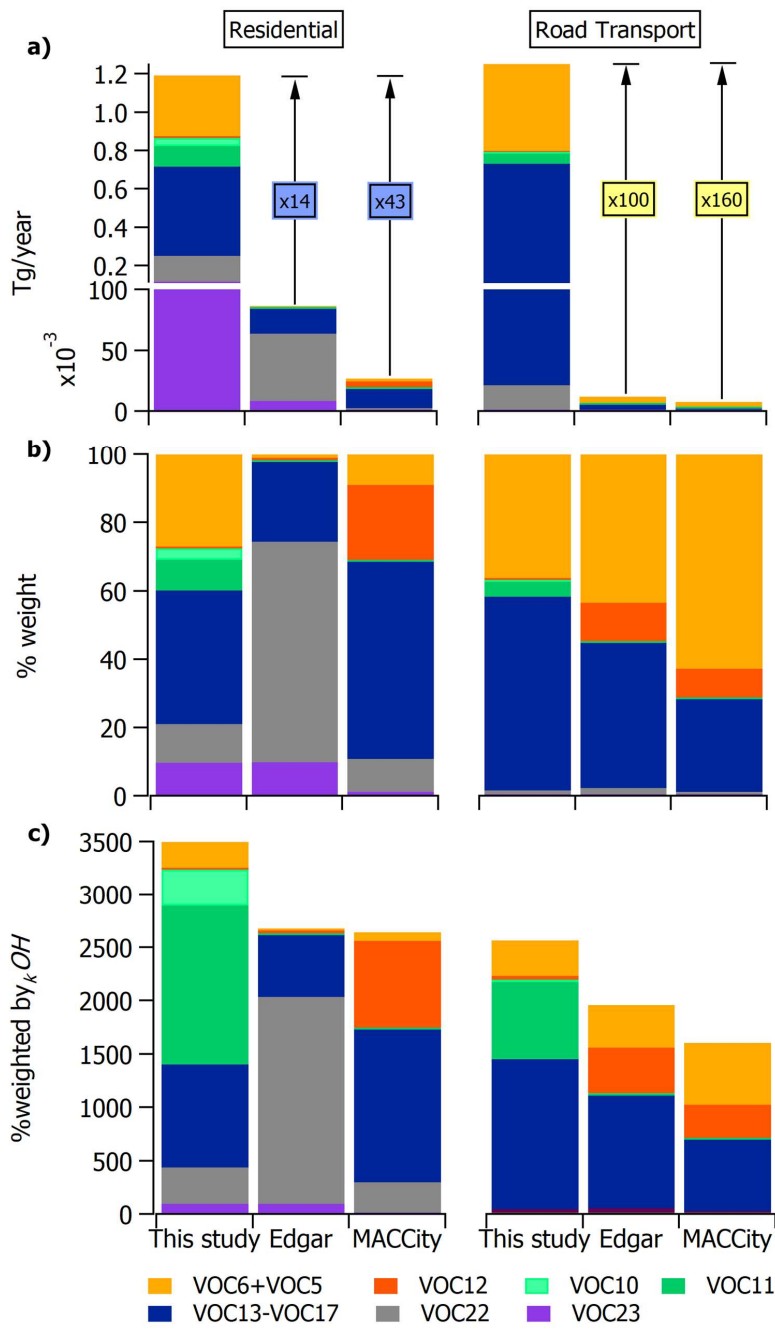

**Figure 8**. Comparison of VOC emission profiles for Côte d'Ivoire between the emissions estimated from the measurements of this study (Keita et al., 2018), Edgar v4.3.2 (Huang et al.; 2017) and MACCity inventory (Granier et al., 2011).The profile analysis integrates road transportation and residential sectors based on the sector activity for 2012. a) Absolute emissions in Tg/year, b) relative mass contribution, and c) relative mass reactivity considering 100 Tg of emissions weighted by the $k$OH reaction rate calculated for each VOC family.





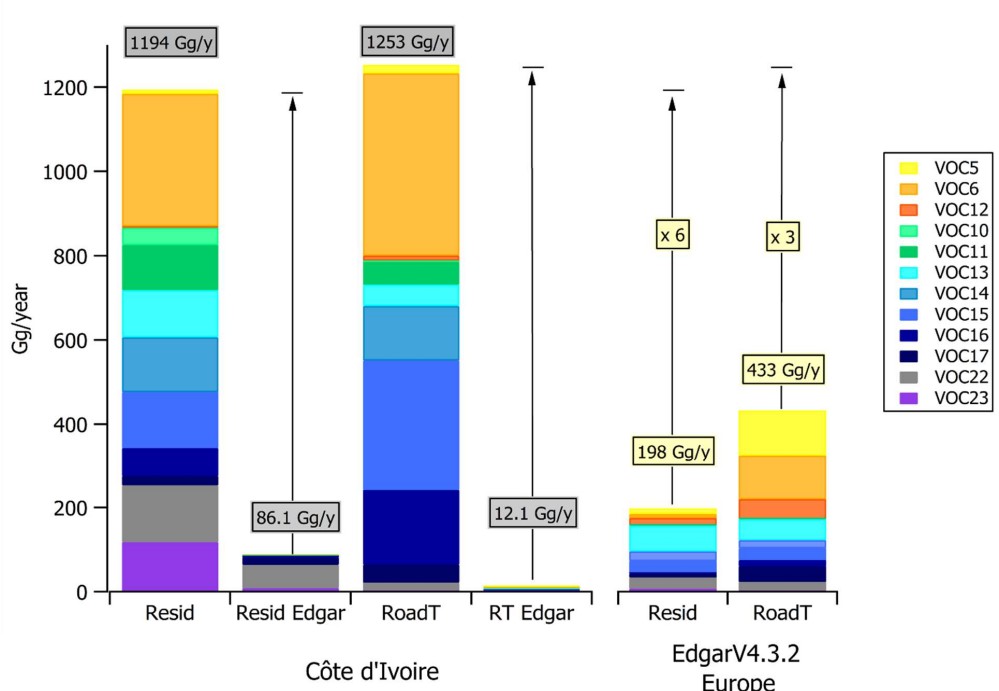


**Figure 9**. VOC emission profiles considering all the VOC families calculated from the measurements of our study and compared with the global Edgar v4.3.2 inventory (Huang et al.; 2017). The comparison integrates road transportation (RoadT and RT) and residential (Resid) sectors in Côte d'Ivoire and Europe for the year 2012. Absolute emissions are expressed in Gg/year for each VOC group.





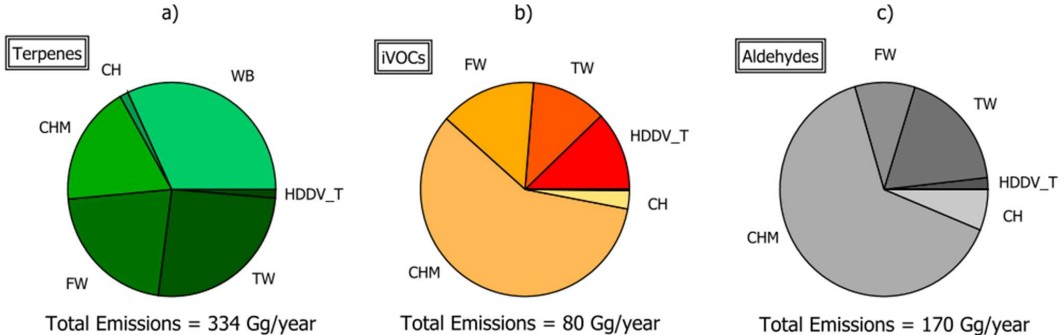

**Figure 10**. Total estimated emissions and relative distributions in the anthropogenic sources measured in Abidjan,
Côte d'Ivoire. Each pie chart represents a VOC family, like a) Terpenes, b) IVOCs and c) Aldehydes.





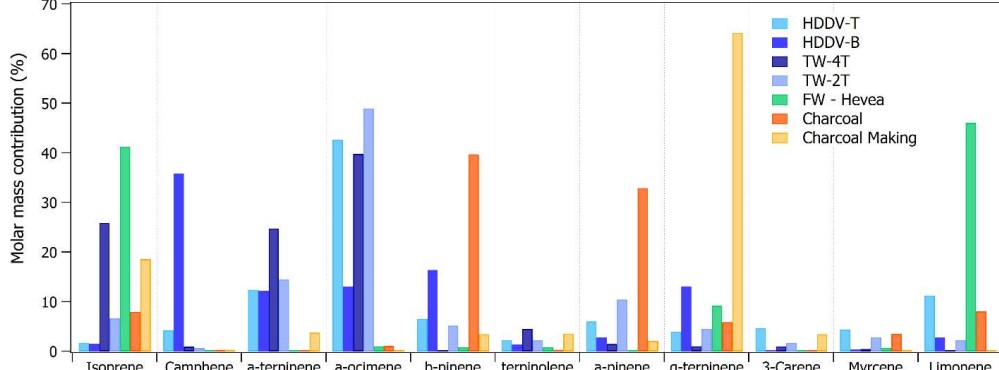

**Figure 11**. Distribution of monoterpenes and isoprene in the emission sources measured in Abidjan. The values represent the percentage of each biogenic VOC over the total emission estimated for these species.




**List of Tables**



Table 1. Description of the emission sources measured and evaluated in Abidjan, Côte d'Ivoire.

| Reference | Sub-group | Description | source | type |
|---|---|---|---|---|
| HDDV | | Heavy-duty diesel vehicles | Diesel emissions | Road Transport |
| | HDDV-T | Diesel trucks | Diesel emissions | Road Transport |
| | HDDV-B | Diesel buses | Diesel emissions | Road Transport |
| LDDV | | Light-duty diesel vehicles | Diesel emissions | Road Transport |
| LDGV | | Light-duty gasoline vehicles | Gasoline emissions | Road Transport |
| TW | TW-2T | Two wheelers two strokes | a mixture of smuggled oil and gasoline | Road Transport |
| | TW-4T | Two wheelers four strokes | a mixture of smuggled oil and gasoline | Road Transport |
| CH | | Charcoal | Charcoal burning | Residential |
| FW | | Fuelwood burning | *Hevea brasiliensis* | Residential |
| CHM | | Charcoal making | Charcoal fabrication | Residential |
| WB | | Waste burning | Domestic landfill burning | Waste burning |



Table 2. Geographical location and characteristics of ambient measurement sites in Abidjan, Côte d'Ivoire

| ID | Site location | Longitude | Latitude | Activity |
|---|---|---|---|---|
| AT | Adjame | 04°01'04"W | 05°21'14"N | Traffic site<br>A site near a transport station; regular traffic jams, ancient public transport vehicles |
| AD | Akouédo | 03°56'16"W | 05°21'12"N | Landfill- waste burning<br>The uncontrolled landfill, continuous waste burning |
| FAC | Cocody | 03°59'27"W | 05°20'42"N | Residential<br>University residence |
| BIN | Bingerville | 03°54'07"W | 05°21'30"N | Urban Background<br>Far from traffic, near to Ebrié Lagoon |
| CRE | Treichville | 04°00'10"W | 05°18'41 "N | Green urban area<br>Near to Ebrié Lagoon; much wind |
| ABO | Abobo | 04°04'10"W | 05°26,0'8,0"N | Traffic + residential<br>Townhall, near to the big market of Abobo. Old communal taxis and minibusses in a crowded crossroad, human activities |
| ZI YOP | Yopougon | 04°04'52"W | 05°22'12"N | Industrial area<br>All type industries (cement, agro-industries, plastic and iron processing, pharmaceutical and cosmetics); heavy-duty vehicles and traffic jams |
| KSI | Koumassi | 03°57'20"W | 05°17'52"N | Domestic fires + traffic<br>A residential site mainly influenced by domestic activities, fire-wood, and charcoal; old vehicles |
| PL | Plateau | 04°01'26"W | 05°19'33"N | Traffic/ administrative<br>City center, crossroad with traffic jams; Light-duty vehicles, near the train station |