# Peer review of "Anthropogenic VOC in Abidjan, southern West Africa: from source"

_Atmospheric Chemistry and Physics, 2018_

## Referee Comment (RC1) · Anonymous Referee #1 · 2 Feb 2019

This paper presents an analysis of "up to 56 VOC" measurements made in Abidjan, Côte d'Ivoire at different ambient sites comprised of different emissions sources using sorbent tubes analyzed on a laboratory GC-FID and GC-MS. This paper is part of the DACCIWA (Dynamics-Aerosol-Chemistry-Cloud Interactions in West Africa) program. Much of the source analysis has already been published in Keita et al., 2018 (https://www.atmos-chem-phys.net/18/7691/2018/), including the source measurement work using sorbent tubes and emission factor calculations for a number of stationary and mobile sources, and I found the line between the previous paper and this paper was very blurred and made it so this paper feels like less of a standalone paper, and more of an addendum to the previous work.

[Figure]

The authors report emission ratios based on measurements made at several different locations in Abidjan meant to correspond with previously-reported source emission factors. It is difficult to fully understand the measurement analysis, however, as the information given in the paper regarding the sampling strategy was very general, and due to a data embargo, no data were made available for the manuscript discussion period. Presumably this data will be made available prior to the finalization of this paper, but I do not feel I can properly assess the paper without more details about the measurements.

My primary issue with this paper, however, is that there are a very large number of errors in typography, grammar and inappropriate word choice such that I find the message of the paper is lost due to these errors. Many of the errors should have been caught by a careful reading and some small attention to detail. I include below some of the basic comments that I have noted, as well as a short list of the technical notes that I made in the first handful of pages and for the figures and tables, but I regret that I am unable to fully assess the science of the paper while these errors exist. For this reason, I recommend that this paper be rewritten and then resubmitted once these typographical, syntax, grammatical, and English language errors are corrected. Please note that the comments and technical notes listed below are by no means a comprehensive list of the issues with the manuscript, as my role is reviewer and not copy editor. I would be happy to review the paper again once it is carefully checked for the above errors and resubmitted.

General comments:

Page 3, line 95: The authors are describing the "main differences. . . associated with the emission source estimations. . .", but are ambiguous about what they're comparing. Are the differences between the inventories and the measurements? Or between inventories? Be specific. Also, the inventory or inventories need to be properly described when they're first discussed.

The sampling strategy for the ambient VOC measurements is not explained well. The authors state that the sampling took place for one month, during which "samples were collected once a week at different daytime." They go on to explain that active sampling of VOCs using a manual pump was carried out on sorbent cartridges... exposed several times a week at each site which corresponds to a total volume of 600 mL. Does this mean that one single cartridge was brought back to the same site and exposed several times over the course of a week or a month? Or was it analyzed between each sampling? Is this described elsewhere? Please detail exactly how many times and at what times of day each cartridge was sampled give a schematic of the sampling mechanism and sampling strategy.

In general, I would prefer the places where ambient sampling took place to be referred to as "sampling locations", rather than "sampling points."

Considering the availability of comprehensive VOC measurement capabilities, it seems inappropriate to suggest that the measurement of 56 VOCs is "extensive", although it is impressive. I just recommend avoiding hyperbole. Further, I do not have the ability to assess the extensiveness of the measurements, because the data are not yet publicly available. This seems out of step with current practices, which generally state that ideally the data for publications be available in an independently-managed DOI. At the very least, the data should be made available at the time of submission.

Lines 349-350: I fail to see how the authors came to this conclusion. There is a lack of information about the proximity of each sampling location to any nearby sources, wind speed and wind direction data, sampling times, etc., and so much of this feels very arbitrary. As well, "the commonalities in spatial distribution seem to be also related..." this is very hand-wavy. Without some regional dispersion modeling detailing the sources, the authors seem to be jumping to conclusions that are not backed up by their measurements or careful analysis.

Technical Notes:

Page 1, line 21: define VOC

Page 1, line 24: Indicate "and later analyzed in a laboratory...", not "the laboratory".

Page 1, line 25: when describing "two-wheelers" in the abstract, please specify that these are two-stroke or four-stroke motorized two-wheelers.

Page 1, line 32: overpassing is likely not the right word here.

Page 1, line 33: insert "organic" into "secondary aerosol formation."

Page 1, line 33: define POCP.

Page 1, line 36: "at the national level"

Page 2, line 40: "For only Côte d'Ivoire..."

Page 2, line 41: "the whole of Europe"

Page 2, line 42: "sectors for Côte d'Ivoire, there is..."

Page 2, lines 45-46: rather than "essential sources", perhaps "widespread" or "ubiquitous"?

Page 2, line 51: "The Western Africa region..."

Page 2, line 61: "... from remote sources, i.e., aerosol dust from..."

Page 2, line 62: "biomass burning plumes and local urban..."

Page 2, line 63: "... campaign showed that air quality..."

Page 3, line 87: "... quantify their emission sources (e.g., Bechara et al., ... ) – this list is not a comprehensive list of all the field campaigns to quantify the VOC in the atmosphere, but rather a small subset of examples.

Page 3, line 94: "for global scales and involve numerous..."

Page 3, line 99: "... inventories commonly estimate (not report?) the total mass of

[Figure]

VOCs.

Page 3, line 101: this sentence needs to be reworded for clarity.

Page 3, line 105: "by a factor of 50..." – for what species or group of species? Be specific.

Page 3, line 110: "their impact on human {what?} and air quality conditions."

Page 4, line 116: "construction of a regional emission..."

Page 4, line 119: "operation as much as possible..."

Page 4, line 126: "sources is assessed (?) regarding... and secondary pollutant formation."

Page 4, line 136: "Health,"

Page 4, line 143: replace susceptible with "responsible?" or another more appropriate word.

Page 4: line 143: replace combine with "include"?

Page 5, line 163: delete the space in "( Keita" – there are many places throughout the paper where spaces are either present or absent in this kind of way. Also: line 167; line 330; line 342; line 426;

Figure 1 caption; Figure 2 caption; Figure 3 caption.

Also throughout: 1) in units, remove decimals (i.e., mL.minˆ-1 should be mL minˆ-1) 2) ppmv and ppbv – the v should not be subscripted. And define ppmv and ppbv (parts per million by volume, parts per billion by volume.)

Page 5, line 172: "Secondly, ambient VOC measurements..."

Page 5, line 175: The selection of sampling locations (not points) were also identified to compare with the emission sources previously measured.

Page 5, 179: "at different times of day."

Page 5: 189: "were performed and analysed…"

Line 287: "in Table SM1, in the..."

Line 295: there are no coordinates in the plot as mentioned.

Line 302: 2119 kmˆ2

Line 303: "In summer, West Africa is influence..."

Line 324: this is not a complete sentence.

Line 342: be consistent with spaces.

Line 343: "Figure 3 shows..."

Line 347: "2018). A similar spatial..."

Line 400: "As already depicted in the previous section, "

Line 402: control doesn't seem like the right word.

Line 427: governed doesn't seem like the right word.

Line 434: The estimation {of what} was based on the calculated...?

Line 444: this is not a complete sentence.

Line 450: "developed by Derwent et al. (2010a)..."

Line 459: is it TW-2T or TW2T?

Figure 2 caption: "Data were downloaded…"

Figure 5: Re: caption: be consistent with hyphens. Probably they're not necessary. The fonts used in the plot are difficult to read. Perhaps use a simpler san serif font. Also, this information and the breakdown of the pie charts shown would be far more

informative in a table.

Figure 8 caption: "(Huang et al., 2017)". And a space is missing after 2011.)

Figure 9 caption: "(Huang et al., 2017)". Also, RT and RoadT seems redundant. Please define better.uj

Figure 10: caption should read: "Total estimated emissions and relative distributions in the... d'Ivoire for the VOC family a) Terpenes, b) IVOCs and c) Aldehydes."

Table 1: "Two wheelers two strokes" doesn't read very well. Perhaps "Two stroke two wheelers"? Also, should "smuggled oil" be defined? Is this different from all other oils? Smuggled is not referred to elsewhere in the manuscript, which makes me intrigued.

Table 2: Why ZI YOP? Why not YOP? The latitude for ABO needs to be corrected (remove commas?) and it would be better if it fit on one line like the other locations. The entire table should be adjusted so that the "activity" can be easily associated with each row – it is difficult to read which lines go with which rows to the left.

Table SM1: Should this just be "S1?" Also, the capitalization of VOCs in the table (and naming) is inconsistent. There are several typos in the info near the bottom. The k in kOH is both italicized and not italicized. Instead of using a "+", perhaps used a superscripted letter and then give the info as a footnote in the table? Why n12 and not n-dodecane? Camphre should be camphor (in English). Why are kOH values being estimated from "analogous species"? Which species? And if some kOH are available, use a different superscript footnote for the kOH values that are estimated from different (specific) species, and also use a different superscript footnote for the SOAP values estimated from different (specific) analogous species. (More information is needed all around.) Again – there are spaces where they don't and lack of spaces where they do belong.
* * *

---

## Referee Comment (RC2) · Anonymous Referee #2 · 12 Mar 2019

This work describes results of VOC analysis of sorbent tube sampling from various regions and sources in and around the SW African coastal city of Abidjan, Côte d'Ivoire. The results are employed to establish fractional molar mass contributions and for the estimation of potential VOC-OH reactivity, ozone and secondary organic aerosol formation.

The emissions factors were compared with those reported in global emission inventories (MACCity and Edgar). The huge emission inventory underestimations reported by this work for speciated VOCs particularly when comparing residential and transportation sectors with the computer model inventory estimations makes a good case for the

need for more such measurements for the larger West African region.

While I am sure that this manuscript contains a lot of novel data that will be of great value to the emissions inventory community, it is difficult to work out how many samples are measured and exactly what is new in this manuscript, rather than what is already covered by other publications, particularly Keita et al., 2018.

It is not until line 545 that I get a better idea of where these new results fit in with those of Keita et al "Our results emphasize the first insights obtained in the work of Keita et al. (2018) . . .) though "emphasize" I feel is not the best word to use... maybe "reinforce"?

I agree with the main points raised by the other reviewer, especially that the quality of the English, is not quite good enough for me to be confident I understand all the points that the authors are trying to make – and it certainly makes for slow reading. Therefore I found it difficult to assess the manuscript in its entirety. I also feel that the manuscript needs to be shortened and "streamlined" to make it more accessible to the reader.

Specific comments (not a comprehensive list):

Line 25: "two-wheeled vehicles" sounds more accurate/scientific/less slang than "two-wheelers".

Line 184 and Table SM1 – how many samples represent each category? What are the standard deviations for each category?

Line 287 I do not see POCP values for each VOC in Table SM1

Table SM2 – I do not see any POCP values on my version of the table.

Line 327 "This analysis relies on the fifteen VOC species already described in Keita et al (2018)" – does this mean that the Keita et al data are used here – or the same chemical species newly measured?
* * *
[Figure]

2018.

---

## Author Comment (AC1) · 16 May 2019

Response to the reviewer's comments We would like to thank the reviewers for their mindful comments on the paper. We have worked hard to comply with all of them. The whole manuscript has been improved, and several changes were introduced in the material and methods, results and discussion and quality of the figures. In the following, the comments made by the referees appear in black, while our replies are in green, and the proposed modified text in the manuscript is in blue.

Anonymous Referee #1 This paper presents an analysis of "up to 56 VOC" measurements made in Abidjan, Côte d'Ivoire at different

ambient sites comprised of different emissions sources using sorbent tubes analyzed on a laboratory GC-FID and GC-MS. This paper is part of the DACCIWA (Dynamics-Aerosol-Chemistry-Cloud Interactions in West Africa) program. Much of the source analysis has already been published in Keita et al., 2018(https://www.atmos-chem-phys.net/18/7691/2018/), including the source measurement work using sorbent tubes and emission factor calculations for a number of stationary and mobile sources, and I found the line between the previous paper and this paper was very blurred and made it so this paper feels like less of a standalone paper, and more of an addendum to the previous work.

The data presented in Keita et al 2018 was obtained under the same work package of the DACCIWA project. In Keita et al.'s paper, only the emission factors of 15 VOC data were described among with particle emission factors. The quantification of VOC emissions was only focused on the road transport sector. In this new paper, the VOC dataset is extended to 56 VOCs including Intermediate Volatile Organic Compounds which were not included in Keita's paper. The measurements not only include emission factors but also ambient mixing ratios in Abidjan. Finally the analysis presents the variability of ambient concentrations, the analysis of the emission factors, the estimation of VOC anthropogenic emissions for all source sectors (not only road transport) from this extended dataset and the evaluation of the atmospheric impacts of the emissions on the regional chemistry.

The authors report emission ratios based on measurements made at several different locations in Abidjan meant to correspond with previously-reported source emission factors. It is difficult to fully understand the measurement analysis, however, as the information given in the paper regarding the sampling strategy was very general, and due to a data embargo, no data were made available for the manuscript discussion period. Presumably this data will be made available prior to the finalization of this paper, but I do not feel I can properly assess the paper without more details about the measurements.

The material and methods section improved are more details are provided for a better understanding of the sampling strategy. Regarding the availability of the data and end of the embargo, currently, all the project data on Sedoo has been moved to a CC-BY license. Since the DACCIWA project did not fully finance VOC data, we have requested their availability in the project website, and they will be soon publicly available.

The ambient campaigns were conducted during the dry season (February 2016). Samples were collected every 2 days at different times of the day (from 6 a.m. to 8 p.m.) by using a manual pump (Accuro 2000, Dräger) at 100 mL sccm flow rate. One single sorbent tube was exposed six times at each sampling location. In total, 3.6 L of air were collected at each site for a single 600 mL-volume each time.

My primary issue with this paper, however, is that there are a very large number of errors in typography, grammar and inappropriate word choice such that I find the message of the paper is lost due to these errors. Many of the errors should have been caught by a careful reading and some small attention to detail. I include below some of the basic comments that I have noted, as well as a short list of the technical notes that I made in the first handful of pages and for the figures and tables, but I regret that I am unable to fully assess the science of the paper while these errors exist. For this reason, I recommend that this paper be rewritten and then resubmitted once these typographical, syntax, grammatical, and English language errors are corrected. Please note that the comments and technical notes listed below are by no means a comprehensive list of the issues with the manuscript, as my role is reviewer and not copy editor. I would be happy to review the paper again once it is carefully checked for the above errors and resubmitted.

We thank the reviewer for all the comments and suggestions. The paper was thoroughly revised by a native speaker and sections have been rewrote for a better understanding of the reader.

General comments: Page 3, line 95: The authors are describing the "main differences.

[Figure]

. . associated with the emission source estimations. . .", but are ambiguous about what they're comparing. Are the differences between the inventories and the measurements? Or between inventories? Be specific. Also, the inventory or inventories need to be properly described when they're first discussed.

We have improved the manuscript quality and the discussion about comparisons.

The sampling strategy for the ambient VOC measurements is not explained well. The authors state that the sampling took place for one month, during which "samples were collected once a week at different daytime." They go on to explain that active sampling of VOCs using a manual pump was carried out on sorbent cartridges. . . exposed several times a week at each site which corresponds to a total volume of 600 mL. Does this mean that one single cartridge was brought back to the same site and exposed several times over the course of a week or a month? Or was it analyzed between each sampling? Is this described elsewhere? Please detail exactly how many times and at what times of day each cartridge was sampled give a schematic of the sampling mechanism and sampling strategy.

Thank you for this valuable remark. The methodology section was rewrote including the details here request.

The ambient campaigns were conducted during the dry season (February 2016). Samples were collected every 2 days at different times of the day (from 6 a.m. to 8 p.m.) by using a manual pump (Accuro 2000, Dräger) at 100 mL sccm flow rate. One single sorbent tube was exposed six times at each sampling location. In total, 3.6 L of air were collected at each site for a single 600 mL-volume each time.

In general, I would prefer the places where ambient sampling took place to be referred to as "sampling locations", rather than "sampling points."

The text was changed as suggested.

Considering the availability of comprehensive VOC measurement capabilities, it seems

inappropriate to suggest that the measurement of 56 VOCs is "extensive", although it is impressive. I just recommend avoiding hyperbole. Further, I do not have the ability to assess the extensiveness of the measurements, because the data are not yet publicly available. This seems out of step with current practices, which generally state that ideally the data for publications be available in an independently-managed DOI. At the very least, the data should be made available at the time of submission.

We agree with the reviewer that extensive is not the more appropriate word. We have changed it by extended dataset.

Lines 349-350: I fail to see how the authors came to this conclusion. There is a lack of information about the proximity of each sampling location to any nearby sources, wind speed and wind direction data, sampling times, etc., and so much of this feels very arbitrary. As well, "the commonalities in spatial distribution seem to be also related..."this is very hand-wavy. Without some regional dispersion modeling detailing the sources, the authors seem to be jumping to conclusions that are not backed up by their measurements or careful analysis.

We thank the reviewer for this suggestion. The results were reanalysed, and the discussion was changed taking into account the reviewer's comments.

Technical Notes:

Page 1, line 21: define VOC Page 1, line 24: Indicate "and later analyzed in a laboratory. . .", not "the laboratory". Page 1, line 25: when describing "two-wheelers" in the abstract, please specify that these are two-stroke or four-stroke motorized two-wheelers. Page 1, line 32: overpassing is likely not the right word here. Page 1, line 33: insert "organic" into "secondary aerosol formation." Page 1, line 33: define POCP. Page 1, line 36: "at the national level" Page 2, line 40: "For only Côte d'Ivoire. . ." Page 2, line 41: "the whole of Europe" Page 2, line 42: "sectors for Côte d'Ivoire, there is. . ." Page 2, lines 45-46: rather than "essential sources", perhaps "widespread" or "ubiquitous"? Page 2, line 51: "The Western Africa region. . ." Page 2, line 61: ". . . from

remote sources, i.e., aerosol dust from. . ." Page 2, line 62: "biomass burning plumes and local urban. . ." Page 2, line 63: ". . . campaign showed that air quality. . ." Page 3, line 87: ". . . quantify their emission sources (e.g., Bechara et al., . . . ) – this list is not a comprehensive list of all the field campaigns to quantify the VOC in the atmosphere, but rather a small subset of examples. Page 3, line 94: "for global scales and involve numerous. . ." Page 3, line 99: ". . . inventories commonly estimate (not report?) the total mass of VOCs. Page 3, line 101: this sentence needs to be reworded for clarity. Page 3, line 105: "by a factor of 50. . ." – for what species or group of species? Be specific. Page 3, line 110: "their impact on human {what?} and air quality conditions." Page 4, line 116: "construction of a regional emission. . ." Page 4, line 119: "operation as much as possible. . ." Page 4, line 126: "sources is assessed (?) regarding. . . and secondary pollutant formation." Page 4, line 136: "Health," Page 4, line 143: replace susceptible with "responsible?" or another more appropriate word. Page 4: line 143: replace combine with "include"? Page 5, line 163: delete the space in "( Keita" – there are many places throughout the paper where spaces are either present or absent in this kind of way. Also: line 167; line 330; line 342; line 426; Figure 1 caption; Figure 2 caption; Figure 3 caption.

Also throughout: 1) in units, remove decimals (i.e., mL.minËȨ-1 should be mL minËȨ-1) 2) ppmv and ppbv – the v should not be subscripted. And define ppmv and ppbv (parts per million by volume, parts per billion by volume.) Page 5, line 172: "Secondly, ambient VOC measurements. . ." Page 5, line 175: The selection of sampling locations (not points) were also identified to compare with the emission sources previously measured.

Page 5, 179: "at different times of day." Page 5: 189: "were performed and analysed. . ." Line 287: "in Table SM1, in the..." Line 295: there are no coordinates in the plot as mentioned. Line 302: 2119 kmËȨ2 Line 303: "In summer, West Africa is influence..." Line 324: this is not a complete sentence. Line 342: be consistent with spaces. Line 343: "Figure 3 shows..." Line 347: "2018). A similar spatial..." Line 400: "As already

depicted in the previous section, " Line 402: control doesn't seem like the right word. Line 427: governed doesn't seem like the right word. Line 434: The estimation {of what} was based on the calculated...? Line 444: this is not a complete sentence. Line 450: "developed by Derwent et al. (2010a)..." Line 459: is it TW-2T or TW2T? Figure 2 caption: "Data were downloaded. . ." Figure 5: Re: caption: be consistent with hyphens. Probably they're not necessary. The fonts used in the plot are difficult to read. Perhaps use a simpler san serif font. Also, this information and the breakdown of the pie charts shown would be far more informative in a table. Figure 8 caption: "(Huang et al., 2017)". And a space is missing after 2011.) Figure 9 caption: "(Huang et al., 2017)". Also, RT and RoadT seems redundant. Please define better.uj Figure 10: caption should read: "Total estimated emissions and relative distributions in the. . . d'Ivoire for the VOC family a) Terpenes, b) IVOCs and c) Aldehydes."

We thank the reviewer for the detailed revision of our manuscript. We have revised the manuscript and changed the typos and errors as suggested.

Table 1: "Two wheelers two strokes" doesn't read very well. Perhaps "Two stroke two wheelers"? Also, should "smuggled oil" be defined? Is this different from all other oils? Smuggled is not referred to elsewhere in the manuscript, which makes me intrigued.

We thank the reviewer for this remark. We have changed the table and text as suggested and also incorporate the information of smuggling oil into the manuscript.

In African countries, two-wheeled vehicles (two-stroke or four-stroke engines) frequently use a mixture of oil and gasoline derived from smuggling, which is characterized by high pollutant emissions (Assamoi and Liousse, 2010).

Table 2: Why ZI YOP? Why not YOP? The latitude for ABO needs to be corrected (remove commas?) and it would be better if it fit on one line like the other locations. The entire table should be adjusted so that the "activity" can be easily associated with each row – it is difficult to read which lines go with which rows to the left.

We have included the suggestions in the table and the typos were corrected.

Table SM1: Should this just be "S1?" Also, the capitalization of VOCs in the table (and naming) is inconsistent. There are several typos in the info near the bottom. The k in kOH is both italicized and not italicized. Instead of using a "+", perhaps used a superscripted letter and then give the info as a footnote in the table? Why n12 and not n-dodecane? Camphre should be camphor (in English). Why are kOH values being estimated from "analogous species"? Which species? And if some kOH are available, use a different superscript footnote for the kOH values that are estimated from different (specific) species, and also use a different superscript footnote for the SOAP values estimated from different (specific) analogous species. (More information is needed all around.) Again – there are spaces where they don't and lack of spaces where they do belong.

We thank the reviewer for these significant suggestions. We have included the suggestions in the table and the typos were corrected. Anonymous Referee #2 This work describes results of VOC analysis of sorbent tube sampling from various regions and sources in and around the SW African coastal city of Abidjan, Côte d'Ivoire. The results are employed to establish fractional molar mass contributions and for the estimation of potential VOC-OH reactivity, ozone and secondary organic aerosol formation. The emissions factors were compared with those reported in global emission inventories (MACCity and Edgar). The huge emission inventory underestimations reported by this work for speciated VOCs particularly when comparing residential and transportation sectors with the computer model inventory estimations makes a good case for the need for more such measurements for the larger West African region.

While I am sure that this manuscript contains a lot of novel data that will be of great value to the emissions inventory community, it is difficult to work out how many samples are measured and exactly what is new in this manuscript, rather than what is already covered by other publications, particularly Keita et al., 2018.

[Figure]

We thank the reviewer for this remark. The manuscript has been thoroughly revised and the material and methods section was rewrote.

It is not until line 545 that I get a better idea of where these new results fit in with those of Keita et al "Our results emphasize the first insights obtained in the work of Keita et al. (2018) : : :) though "emphasize" I feel is not the best word to use... maybe "reinforce"?

The text was changed as suggested.

I agree with the main points raised by the other reviewer, especially that the quality of the English, is not quite good enough for me to be confident I understand all the points that the authors are trying to make – and it certainly makes for slow reading. Therefore I found it difficult to assess the manuscript in its entirety. I also feel that the manuscript needs to be shortened and "streamlined" to make it more accessible to the reader.

We thank the reviewer for the time and effort put in the revision of the manuscript. The paper was thoroughly revised by a native speaker and sections have been rewrote for a better understanding of the reader.

Specific comments (not a comprehensive list): Line 25: "two-wheeled vehicles" sounds more accurate/scientific/less slang than "twowheelers". The text was changed by two-wheel vehicles

Line 184 and Table SM1 – how many samples represent each category? What are the standard deviations for each category?

We have incorporated the information requested in the supplementary material and in the Sampling section

Line 287 I do not see POCP values for each VOC in Table SM1 Thank you for the comment. We have used the POCP values related to each VOC family as it was reported in the work of Huang et al., 2017. Thus, we finally did not incorporate the individual POCP values in the Table S1

Table SM2 – I do not see any POCP values on my version of the table. Thank you for this remark, the values were incorporated in the Table S2

Line 327 "This analysis relies on the fifteen VOC species already described in Keita et al (2018)" – does this mean that the Keita et al data are used here – or the same chemical species newly measured?

The data presented in Keita et al 2018 was obtained under the same work package of the DACCIWA project. Despite we integrate here the fifteen VOC compounds already assessed in Keita et al., we present them in a deeper analysis, not only by analysing their emission factors but also by evaluating the potential impacts on the regional atmospheric chemistry

Please also note the supplement to this comment:
https://www.atmos-chem-phys-discuss.net/acp-2018-1263/acp-2018-1263-AC1-supplement.pdf

---

## Author Comment (AC3) · 16 May 2019

Supplementary material of

**Anthropogenic VOC in Abidjan, southern West Africa: from source quantification to atmospheric impacts**

*Dominutti P., et al (pamela.dominutti@york.ac.uk)*

Table S1. Mean VOC to CO emission ratios (ERs, ppbv per ppmv CO) for each emission source analysed in this study.

| | MW (g mol$^{-1}$) | SOAP value | kOH value | Mean ER VOC/CO ± SD (ppbv/ppmv) | | | | | | | | | |
|---|---|---|---|---|---|---|---|---|---|---|---|---|---|
| | | | | HDDV | TW 2T | TW 4T | CH | FW | CHM | WB | LDDV | LDGV | HDDV-T |
| benzene | 78.11 | 92.9 | 1.2 | 0.42 ± 0.03 | 53.2 ± 10.7 | 10.5 ± 4.17 | 1.76 ± 1.80 | 1.51 | 0.94 | 12.75 ± 8.43 | 3.06 ± 1.17 | 1.41 ± 0.25 | 13.74 |
| toluene | 92.14 | 100 | 5.6 | 0.26 ± 0.05 | 117.6 ± 25 | 15.9 ± 4.33 | 0.58 ± 0.43 | 0.83 | 0.67 | 18.1 ± 14.6 | 1.21 ± 0.57 | 6.90 ± 3.16 | 4.62 |
| m+p-xylene | 106.16 | 76.3 | 19 | 0.21 ± 0.04 | 102.1 ± 10 | 8.32 ± 3.77 | 0.32 ± 0.18 | 0.58 | 0.27 | 1.96 ± 2.22 | 0.33 ± 0.04 | 10.0 ± 0.86 | 3.26 |
| o-xylene | 106.16 | 95.5 | 14 | 0.08 ± 0.02 | 58.3 ± 15.7 | 3.22 ± 1.32 | 0.08 ± 0.06 | 0.06 | 0.08 | 0.03 ± 0.04 | 0.17 ± 0.05 | 4.82 ± 1.03 | 1.08 |
| ethylbenzene | 106.17 | 111.6 | 7.5 | 0.07 ± 0.02 | 59.5 ± 15.9 | 5.57 ± 2.55 | 0.29 ± 0.32 | 0.10 | 0.09 | 11.66 ± 8.3 | 0.25 ± 0.08 | 2.53 ± 0.75 | 1.26 |
| styrene | 104.15 | 212.3 | 43 | 0.06 ± 0.01 | 37.5 ± 23.8 | 2.79 ± 2.03 | 2.79 ± 0.09 | 0.16 | 0.16 | 0.00 ± 0.00 | 0.00 ± 0.00 | 0.00 ± 0.00 | 0.05 |
| iso-propylbenzene | 120.19 | 95.5 | 6.6 | 0.01 ± 0.01 | 8.32 ± 1.87 | 0.62 ± 0.43 | n.d. | 0.01 | 0.01 | 0.00 ± 0.00 | 0.00 ± 0.00 | 0.00 ± 0.00 | 0.44 |
| 1,3,5-trimethylbenzene | 120.19 | 13.5 | 60 | 0.06 ± 0.01 | 30.1 ± 13.0 | 0.83 ± 0.35 | 0.00 ± 0.00 | 0.02 | 0.04 | 0.56 ± 0.57 | 0.13 ± 0.01 | 1.32 ± 0.07 | 0.90 |
| 1,2,4-trimethylbenzene | 120.19 | 20.6 | 32 | 0.19 ± 0.04 | 65.6 ± 24.5 | 2.83 ± 1.25 | 0.02 ± 0.01 | 0.02 | 0.10 | 0.49 ± 0.41 | 0.42 ± 0.02 | 4.29 ± 0.54 | 4.17 |
| 1,2,3-trimethylbenzene | 120.19 | 43.9 | 29 | 0.07 ± 0.01 | 20.8 ± 7.58 | 0.62 ± 0.27 | 0.65 ± 0.85 | 0.21 | 0.14 | 0.16 ± 0.15 | 0.17 ± 0.01 | 0.73 ± 0.12 | 1.64 |
| isoprene | 68.12 | 1.9 | 100 | 0.02 ± 0.01 | 3.99 ± 1.88 | 0.68 ± 0.24 | 0.07 ± 0.04 | 1.68 | 0.22 | 2.08 ± 2.84 | 0.04 ± 0.04 | 0.17 ± 0.15 | 0.29 |
| hexene | 84.16 | 0 | 37 | n.d. | 22.7 ± 6.44 | 0.35 ± 0.28 | 0.35 ± 0.11 | 0.00 | 0.00 | 0.00 ± 0.00 | 0.00 ± 0.00 | 0.00 ± 0.00 | 1.06 |
| pentane | 72.15 | 0.3 | 3.8 | 0.01 ± 0.00 | 42.6 ± 8.85 | 3.50 ± 0.55 | 3.50 ± 0.13 | 0.07 | 0.07 | n.d. | n.d. | n.d. | 0.09 |
| 2-methylpentane | 86.18 | 0 | 5.2 | 0.02 ± 0.01 | 73.5 ± 0.43 | 5.91 ± 4.28 | 5.91 ± 2.26 | 0.00 | 0.00 | n.d. | n.d. | n.d. | 0.49 |
| 3-methylpentane[2] | 86.18 | 0.2 | 5.2 | 0.01 ± 0.00 | 61.2 ± 14.9 | 3.53 ± 2.48 | 3.53 ± 3.44 | 0.42 | 0.42 | n.d. | n.d. | n.d. | 0.00 |
| hexane | 86.18 | 0.1 | 5.2 | 0.02 ± 0.00 | 75.1 ± 3.90 | 5.82 ± 4.21 | 5.82 ± 1.21 | 0.01 | 0.01 | n.d. | n.d. | n.d. | 0.06 |
| 2,2-dimethylpentane | 100.21 | 0 | 4.77 | 0.00 ± 0.00 | 7.21 ± 1.95 | 0.12 ± 0.09 | 0.12 ± 0.01 | 0.16 | 0.16 | n.d. | n.d. | n.d. | 0.43 |
| 2,4-dimethylpentane[1,2] | 100.21 | 0.3 | 4.77 | 0.00 ± 0.00 | 24.3 ± 2.70 | 0.38 ± 0.27 | 0.38 ± 0.02 | 0.02 | 0.02 | n.d. | n.d. | n.d. | 0.00 |
| 2,2,3-trimethylbutane[1] | 100.21 | 0.3 | 3.81 | 0.00 ± 0.00 | 3.07 ± 0.89 | 0.06 ± 0.05 | 0.06 ± 0.01 | 0.02 | 0.02 | n.d. | n.d. | n.d. | 0.06 |
| 3,3-dimethylpentane[1,2] | 100.21 | 0.3 | 4.77 | 0.01 ± 0.00 | 8.45 ± 1.75 | 0.02 ± 0.01 | 0.02 ± 0.00 | 0.21 | 0.21 | n.d. | n.d. | n.d. | 0.13 |
| cyclohexane | 84.16 | 1 | 7 | 0.00 ± 0.00 | 26.6 ± 15.8 | 1.66 ± 1.09 | 1.66 ± 0.35 | 0.01 | 0.01 | n.d. | n.d. | n.d. | 0.05 |
| 2-methylhexane[2] | 100.21 | 0 | 7 | 0.01 ± 0.00 | 68.4 ± 14.4 | 1.81 ± 1.31 | 1.81 ± 0.00 | 0.05 | 0.05 | n.d. | n.d. | n.d. | 0.17 |
| 2,3-dimethylpentane[2] | 100.21 | 0.4 | 4.77 | n.d. | 20.6 ± 10.9 | 0.55 ± 0.40 | 0.55 ± 0.00 | 0.00 | 0.00 | n.d. | n.d. | n.d. | 0.01 |
| heptane | 100.21 | 0.1 | 6.8 | 0.13 ± 0.07 | 40.0 ± 13.1 | 2.34 ± 1.17 | 0.09 ± 0.05 | 0.02 | 0.53 | 3.09 ± 3.40 | 0.06 ± 0.04 | 0.08 ± 0.04 | 0.29 |
| octane | 114.23 | 0.8 | 8.1 | 0.25 ± 0.12 | 32.1 ± 13.1 | 0.94 ± 0.48 | 0.10 ± 0.05 | 0.84 | 0.20 | 2.02 ± 1.95 | 0.05 ± 0.02 | 0.33 ± 0.09 | 0.89 |
| iso-octane | 114.23 | 0.8 | 3.34 | 0.03 ± 0.02 | 16.8 ± 6.49 | 0.15 ± 0.05 | 0.04 ± 0.04 | 0.13 | 0.01 | 0.12 ± 0.13 | 0.05 ± 0.03 | 0.01 ± 0.00 | 0.25 |
| nonane | 128.26 | 1.9 | 9.7 | 0.14 ± 0.03 | 10.6 ± 7.36 | 0.32 ± 0.23 | 0.32 ± 0.00 | 0.02 | 0.02 | n.d. | n.d. | n.d. | 3.71 |
| decane | 142.29 | 7 | 11 | 0.24 ± 0.06 | 6.66 ± 3.49 | 0.15 ± 0.10 | 0.15 ± 0.02 | 0.00 | 0.00 | n.d. | n.d. | n.d. | 8.26 |

| | MW (g mol[-1]) | SOAP value | kOH value | Mean ER VOC/CO ± SD (ppbv/ppmv) | | | | | | | | | |
|---|---|---|---|---|---|---|---|---|---|---|---|---|---|
| | | | | HDDV | TW 2T | TW 4T | CH | FW | CHM | WB | LDDV | LDGV | HDDV-T |
| undecane | 156.31 | 16.2 | 12 | 0.28 ± 0.02 | 2.05 ± 1.71 | 0.06 ± 0.04 | 0.06 ± 0.05 | 0.00 | 0.00 | n.d. | n.d. | n.d. | 0.62 |
| dodecane | 170.33 | 34.5 | 13.2 | 0.28 ± 0.02 | 1.01 ± 1.05 | 0.03 ± 0.02 | 0.03 ± 0.00 | 0.01 | 0.01 | n.d. | n.d. | n.d. | 14.72 |
| tridecane[1] | 184.36 | 34.5 | 15.1 | 0.02 ± 0.00 | 0.08 ± 0.05 | 0.00 ± 0.00 | 0.00 ± 0.00 | 0.00 | 0.00 | n.d. | n.d. | n.d. | 0.17 |
| tetradecane[1] | 198.39 | 34.5 | 17.9 | 0.06 ± 0.01 | 0.24 ± 0.18 | 0.01 ± 0.00 | 0.01 ± 0.00 | 0.01 | 0.01 | n.d. | n.d. | n.d. | 2.19 |
| pentadecane[1] | 212.41 | 34.5 | 20.7 | 0.02 ± 0.00 | 0.08 ± 0.08 | 0.00 ± 0.00 | 0.00 ± 0.00 | 0.04 | 0.04 | n.d. | n.d. | n.d. | 1.23 |
| hexadecane[1] | 226.44 | 34.5 | 23.2 | 0.02 ± 0.00 | 0.08 ± 0.03 | 0.00 ± 0.00 | 0.00 ± 0.00 | 0.01 | 0.01 | n.d. | n.d. | n.d. | 0.85 |
| limonene | 136.24 | 18 | 164 | 0.01 ± 0.00 | 0.41 ± 0.30 | 0.00 ± 0.00 | 0.02 ± 0.02 | 0.47 | 0.03 | 10.83 ± 7.9 | 0.00 ± 0.00 | 0.00 ± 0.00 | 0.49 |
| α-pinene | 136.23 | 17.4 | 52.3 | 0.01 ± 0.00 | 1.81 ± 0.73 | 0.01 ± 0.01 | 0.05 ± 0.07 | 0.00 | 0.00 | 0.02 ± 0.02 | 0.00 ± 0.00 | 0.00 ± 0.00 | 0.26 |
| β-pinene | 136.23 | 18.1 | 74.3 | 0.02 ± 0.01 | 0.93 ± 0.59 | 0.01 ± 0.00 | 0.06 ± 0.09 | 0.01 | 0.00 | 0.20 ± 0.19 | 0.00 ± 0.00 | 0.00 ± 0.00 | 0.28 |
| camphene[1] | 136.24 | 18 | 53 | 0.06 ± 0.02 | 0.34 ± 0.31 | 0.01 ± 0.01 | 0.01 ± 0.00 | 0.00 | 0.00 | n.d. | n.d. | n.d. | 0.36 |
| myrcene[1] | 136.23 | 18 | 215 | 0.00 ± 0.00 | 0.65 ± 0.41 | 0.00 ± 0.00 | 0.00 ± 0.01 | 0.01 | 0.01 | n.d. | n.d. | n.d. | 0.37 |
| 3-carene[1] | 136.24 | 18 | 85 | 0.00 ± 0.00 | 0.41 ± 0.45 | 0.01 ± 0.01 | 0.01 ± n.d. | 0.00 | 0.00 | n.d. | n.d. | n.d. | 0.39 |
| α-terpinene[1] | 136.23 | 18 | 363 | 0.02 ± 0.01 | 4.15 ± 2.45 | 0.19 ± 0.13 | 0.19 ± 0.00 | 0.00 | 0.00 | n.d. | n.d. | n.d. | 1.05 |
| α-ocimene[1] | 136.23 | 18 | 252 | 0.02 ± 0.00 | 12.3 ± 4.10 | 0.31 ± 0.17 | 0.31 ± 0.00 | 0.01 | 0.01 | n.d. | n.d. | n.d. | 3.62 |
| γ-terpinene[1] | 136.23 | 18 | 177 | 0.05 ± 0.00 | 0.96 ± 0.75 | 0.01 ± 0.01 | 0.01 ± 0.00 | 0.09 | 0.09 | n.d. | n.d. | n.d. | 0.33 |
| terpinolene[1] | 136.24 | 18 | 225 | 0.00 ± 0.00 | 0.62 ± 0.35 | 0.03 ± 0.03 | 0.03 ± 0.00 | 0.01 | 0.01 | n.d. | n.d. | n.d. | 0.19 |
| hexanal | 100.16 | 0 | 30 | 0.00 ± 0.00 | 14.2 ± 14.2 | 0.03 ± 0.03 | 0.03 ± 0.00 | 0.05 | 0.05 | n.d. | n.d. | n.d. | 0.14 |
| heptanal[1,2] | 114.19 | 0 | 30 | 0.01 ± 0.00 | n.d. | n.d. | n.d. | 0.01 | 0.01 | n.d. | n.d. | n.d. | 0.60 |
| benzaldehyde | 106.12 | 216.1 | 12 | 0.03 ± 0.03 | 5.78 ± 4.43 | 0.10 ± 0.08 | 0.10 ± n.d. | 0.00 | 0.00 | n.d. | n.d. | n.d. | 0.01 |
| octanal[1,2] | 128.21 | 0 | 30 | 0.03 ± 0.01 | n.d. | 0.13 ± 0.09 | 0.13 ± 0.11 | 0.04 | 0.04 | n.d. | n.d. | n.d. | 1.07 |
| nonanal[1,2] | 142.24 | 0 | 30 | 0.01 ± 0.01 | 1.39 ± 0.49 | 0.12 ± 0.10 | 0.12 ± 0.26 | 0.01 | 0.01 | n.d. | n.d. | n.d. | 0.16 |
| nopinone[1] | 138.21 | 18 | 15 | 0.00 ± 0.00 | 0.47 ± 0.39 | 0.03 ± 0.02 | 0.03 ± 0.00 | 0.00 | 0.00 | n.d. | n.d. | n.d. | 1.46 |
| camphor[1] | 152.23 | 18 | 4.3 | 0.05 ± 0.00 | 8.86 ± 4.89 | 0.20 ± 0.15 | 0.20 ± n.d. | 0.07 | 0.07 | n.d. | n.d. | n.d. | 0.02 |
| borneol[1] | 154.25 | 18 | 49 | 0.00 ± 0.00 | 2.31 ± 1.34 | 0.04 ± 0.04 | 0.04 ± 0.00 | 0.00 | 0.00 | n.d. | n.d. | n.d. | 1.78 |
| decanal[1,2] | 156.20 | 0 | 30 | 0.03 ± 0.00 | 3.55 ± 2.83 | 0.11 ± 0.09 | 0.11 ± 0.05 | 0.03 | 0.03 | n.d. | n.d. | n.d. | 1.75 |
| undecanal[1,2] | 170.30 | 0 | 30 | 0.06 ± 0.01 | 0.36 ± 0.14 | 0.01 ± 0.01 | 0.01 ± 0.03 | 0.05 | 0.05 | n.d. | n.d. | n.d. | 4.31 |
| methylethylketone | 72.11 | 0.6 | 1.2 | 0.05 ± 0.02 | n.d. | 0.02 ± 0.01 | 0.02 ± 0.10 | 0.19 | 0.19 | n.d. | n.d. | n.d. | 0.65 |
| methylvinylketone | 70.09 | 1 | 19 | 0.05 ± 0.02 | 1.36 ± 0.67 | 0.05 ± 0.02 | 0.05 ± 0.06 | 0.00 | 0.00 | n.d. | n.d. | n.d. | 1.07 |

MW = molecular weight (g mol[-1]); ER = emission ratio in units of ppbv per ppmv CO equivalent to mmol per mol CO, kOH = second-order reaction rate coefficients of VOC+ OH reaction ($\times 10^{12}$ cm[3] molec[-1] s[-1]) obtained from Manion et al., (2015); SOAP = secondary organic aerosol potential values reported in Derwent et al (2010b). [1] denotes species whose SOAP values were estimated from the analogous species and [2] kOH values were estimated from the analogous species.

Table S2: VOC species groups based on GEIA method and mean POCP values suggested for each family (Huang et al., 2017)

| Class | VOCs families | VOC species integrated in this study | Mean POCP |
|---|---|---|---|
| VOC1 | Alkanols (alcohols) | n.d. | 34.92 |
| VOC2 | Ethane | n.d. | 12.3 |
| VOC3 | Propane | n.d. | 22.12 |
| VOC4 | Butanes | n.d. | 36.54 |
| VOC5 | Pentanes | pentane | 39.5 |
| VOC6 | Hexanes and higher alkanes | 2-methylpentane, 3-methylpentane, hexane, 2,2-dimethylpentane, 2,4-dimethylpentane, 2,2,3-trimethylbutane, 3,3-dimethylpentane, cyclohexane, 2-methylhexane, 2,3-dimethylpentane, iso-octane, heptane, octane, nonane, decane, undecane, dodecane, tridecane, tetradecane, pentadecane, hexadecane | 44.15 |
| VOC7 | Ethene (ethylene) | n.d. | 100 |
| VOC8 | Propene | n.d. | 97.89 |
| VOC9 | Ethyne (acetylene) | n.d. | 8.5 |
| VOC10 | Isoprenes | isoprene | 109.2 |
| VOC11 | Monoterpenes | $\alpha$-pinene, $\beta$-pinene, camphene, 3-carene, $\alpha$-terpinene, limonene, $\alpha$-ocimene, $\gamma$-terpinene, terpinolene, myrcene | 109.2 |
| VOC12 | Other alk(adi)enes/alkynes (olefines) | n.d. | 95.29 |
| VOC13 | Benzene (benzol) | benzene | 21.8 |
| VOC14 | Methylbenzene (toluene) | toluene | 63.7 |
| VOC15 | Dimethylbenzenes (xylenes) | m+p-xylene, o-xylene, ethylbenzene, | 107.41 |
| VOC16 | Trimethylbenzenes | 1,2,4-trimethylbenzene, 1,2,3-trimethylbenzene and 1,3,5-trimethylbenzene | 129.86 |
| VOC17 | Other aromatics | iso-propylbenzene, styrene | 77.78 |
| VOC18 | Esters | n.d. | 20.68 |
| VOC19 | Ethers (alkoxy alkanes) | n.d. | 12.44 |
| VOC20 | Chlorinated hydrocarbons | n.d. | 23.72 |
| VOC21 | Methanal (formaldehyde) | n.d. | 51.9 |
| VOC22 | Other alkanals (aldehydes) | benzaldehyde, heptanal, hexanal, octanal, nonanal, decanal, undecanal, camphor, borneol | 64.1 |
| VOC23 | Alkanones (ketones) | methylvinylketone, methylethylketone | 24.54 |
| VOC24 | Acids (alkanoic) | n.d. | 12.44 |
| VOC25 | Other NMVOC (HCFCs, nitriles, etc.) | n.d. | 12.44 |

POCP= photochemical ozone creation potentials (POCPs) reported on Derwent et al ( 2001, 2010)

**References**

Derwent, R. G., Jenkin, M. E., Saunders, S. M. and Pilling, M. J.: Characterization of the reactivities of volatile organic compounds using a master chemical mechanism, J. Air Waste Manag. Assoc., 51(5), 699–707, doi:10.1080/10473289.2001.10464297, 2001.

Derwent, R. G., Jenkin, M. E., Pilling, M. J., Carter, W. P. L. and Kaduwela, A.: Reactivity scales as comparative tools for chemical mechanisms., J. Air Waste Manag. Assoc., 60(8), 914–924, doi:10.3155/1047-3289.60.8.914, 2010a.

Derwent, R. G., Jenkin, M. E., Utembe, S. R., Shallcross, D. E., Murrells, T. P. and Passant, N. R.: Secondary organic aerosol formation from a large number of reactive man-made organic compounds, Sci. Total Environ., 408(16), 3374–3381, doi:10.1016/j.scitotenv.2010.04.013, 2010b.

Huang, G., Brook, R., Crippa, M., Janssens-Maenhout, G., Schieberle, C., Dore, C., Guizzardi, D., Muntean, M., Schaaf, E. and Friedrich, R.: Speciation of anthropogenic emissions of non-methane volatile organic compounds: A global gridded data set for 1970-2012, Atmos. Chem. Phys., 17(12), 7683–7701, doi:10.5194/acp-17-7683-2017, 2017.

Manion, J. A., Huie, R. E., Levin, R. D., Jr., D. R. B., Orkin, V. L., Tsang, W., McGivern, W. S., Hudgens, J. W., Knyazev, V. D., Atkinson, D. B., Chai, E., Tereza, A. M., Lin, C.-Y., Allison, T. C., Mallard, W. G., Westley, F., Herron, J. T., R. F. Hampson, A. and Frizzell, D. H.: NIST Chemical Kinetics Database, Gaithersburg, Maryland. [online] Available from: http://kinetics.nist.gov/ (Accessed 18 April 2018), 2015.

---

## Author Response (AR2)

**Response to the reviewer's comments**

We would like to thank again the reviewers for their comments and suggestions on the paper.

In the following, the comments made by the referees appear in black, while our replies are in blue, as well as the the proposed *modified text in the manuscript*. We have also considered and changed the technical notes and typos suggested by the reviewers.

General Comments:

This is my second review of this paper, and the authors have addressed many of my concerns from the first round of reviews, although I would have preferred that the "response to reviewers" included significantly more detail to make it easier to see what changes were made between the original submission and this revised version. Nevertheless, many of the grammatical, technical, and word choice issues have been addressed, and have made the paper somewhat more readable, although there remain a long list of technical issues that I have listed at the end of this review since the authors failed to extrapolate my suggestions to the remainder of the paper.

This paper is a continuation of the Keita et al. (2018) paper, a paper that is referenced frequently in this work, which reports emission factors for a subset of measured VOCs Abidjan. There is scientific value in the observations and analyses presented in this paper, and in the connection of the VOC measurements and source emissions in an urban West Africa city to contribute to and improve global emission inventories, and as such, it would be good to see this research published. Nevertheless, this paper still has several issues that need to be addressed prior to publication, and I outline them below.

Finally, I wish to voice my concern again that the data being discussed and analyzed in this paper is still not available. The authors have provided a website, but as the data are not public, it is not possible for me to assess the calculations that they have described in this paper. My recommendation is that the paper not be published until the data are made available to the reviewers, either at the website provided, or at a secondary temporary data repository.

Regarding the major concern on data availability, the author would like to inform the editor and the reviewer that the data are now available on the BAOBAB database http://baobab.sedoo.fr/DACCIWA/News/. These data are the mass concentrations of VOCs in ambient air and at emission. The LaMP and IMT Douai groups were in charge in the analysis of the samples.

Specific Comments:

Line 66: Please reference where the AMMA campaign took place (city and country?)

Dominutti et al.,: AMMA covered different countries and target cities in West Africa, the major ones from an emission perspective being the megacity of Lagos.

Line 97: It is unclear what is meant by "sensitive places with high anthropogenic pressures". Please use a more technical description.

Dominutti et al.,: : the term "sensitive" might be ambiguous and we changed it by the following: "*in places of the developing world with high anthropogenic pressure*"

Line 102: The authors state that global emission inventories commonly estimate the total mass of VOCS. This needs to be explained or corrected. Many inventories include speciation or at the very least, parameterization of various VOCs or VOC groups, and do not just provide a total VOC mass. The authors go on to suggest that the fate and contribution of each species can change depending on the emission source, and again – this needs to be explained somewhat. The fate of the VOC should not depend on the emission source, unless the reference implies that \*other\* co-emitted species could determine the lifetime and ultimate fate of the VOC. The source, however, doesn't not necessarily define the atmospheric fate of all VOCs.

Dominutti et al.,: we agree with the reviewer's suggestions and the target section have been modified. We propose the following (lines 101-105):

*[…] While global emission inventories commonly estimate the total mass of speciated VOCs, they fail in reproducing the spatial and temporal variability of VOC emission speciation. Indeed, the emission composition can change depending on the emission source, fuel quality, combustion technologies, and main regional practices (Huang et al., 2017). The use of activity data and emission factors derived from local measurements of regional-specific sources may help to reduce the uncertainties in those emission inventories. […]*

Line 106: Give a better description of the "different compounds" for which Keita et al. reported EFs. Also, perhaps start line 107 "In that study, a comparison of the emissions calculated from the EFs…" to clarify that this isn't being presented here. Similarly, on line 109, "That study emphasized the importance…" to differentiate work that was presented in another paper from work being presented in this paper.

Dominutti et al.: The manuscript was changed as suggested (lines 106-109).
*[…]A comparison of the emissions calculated from the EFs with those observed from the EDGARv4.3.2 (Huang et al., 2017) inventory showed a marked discrepancy (factor of 50 difference) for fifteen VOCs species (3 alkanes, 8 aromatics, isoprene and 3 monoterpenes) in Côte d'Ivoire. That study emphasised the importance of considering African anthropogenic emissions at regional scales […].*
Regarding the differences between our work and the one from Keita et al, more details can be found also in the lines 174-176.

Line 116: A second Knippertz et al., 2015 seems to be missing from the paper (titled: The DACCIWA Project: Dynamics–Aerosol–Chemistry–Cloud Interactions in West Africa, https://doi.org/10.1175/BAMS-D-14-00108.1). Clarify this, and where necessary, edit all references to Knippertz et al., 2015a or Knippertz et al., 2015b.

Dominutti et al.: we agree with reviewer's suggestion and the reference was included in the manuscript.

Lines 139-150 – This paragraph is not a material or a method. It should be moved to the intro.

Dominutti et al.: as suggested by the editor we moved this part into the introduction (lines 107-119)

Section 2.1 Sampling: This section still lacks the clarity and specifics needed to describe how the sampling was performed. The authors need to consider that the reader does not know how the sampling took place and be very explicit. At the very least, a table indicating the number of samples, (including resampling of cartridges), and times for the sampling for each cartridge, or pair of cartridges would be helpful. As well, the campaign needs to be described in terms of duration, times, intensives, etc. Also, describe the preparation of the cartridges *before* a discussion of the sampling, and the results.

> Dominutti et al.: some of the suggestions have been already taken into account in the first revised version like the preparation of the cartridges which is described at the beginning of the section. For the sampling strategy, we also provided some details in the first revised version. To be more explicit, we added two tables (Table S3 and S4) in the annex summarizing all the experiences carried out to address the spatial variability of VOC mixing ratios and their direct emission measurements. These tables especially report the number of samples, the sampling dates and times and the sampling volume. We hope that these new tables will be explicit enough and fills the gaps on the sampling strategy.

Also, for example, on Line 190: 3 samples on Tenax and Carbopack tubes each? Or 3 samples total? One Tenax and two Carbopack? This is why the sampling should be more explicit. These should be described fully. Which were sampled where? Simultaneously, or back to back?

> Dominutti et al.: More details have been introduced in the sampling strategy section for the emission sources. (lines 156-158). When Tenax and Carbopack tubes were sampled, the measurements were made with both tubes one after another at the same emission source. More details about emission source measurements can be found as well in the new Table S3.

How many samples from each location were used in the summary table S1? What kind of uncertainty is introduced from having samples taken at different times of day? Were there some samples that were higher than others? Were there times of time that were more concentrated than others? Were the samples taken sequentially over a few days, and then new samples taken, or were samples taken only at specific times of day and then other cartridges sampled at other times of day?

> Dominutti et al.: Table S1 is related to emission sources and not to sampling locations. The sampling strategy used for the ambient measurements is now detailed in the Table S3. In order to have a temporally integrated sample at each location to address the spatial variability, the sampling was carried out 6 times over different times of the day (from early morning to evening) and dates, at each location, on the same sorbent tube and with the same volume (600 mL). Two tubes in parallel (almost) were deployed at each location. At the end we obtained two samples at each location which integrate 6 sampling periods with a total air volume of 3.6L. More details about time can be now found in Table S3. Taking samples at different times of the day prevents from the bias introduced by major factors controlling concentrations (emissions, chemistry and dilution).

With no access to the data, I have no idea how many samples were taken at each location, when they were taken, and whether there is any bias introduced into the sampling strategy as a result. This is one of my primary issues with the paper.

Dominutti et al.: The data is now available on the BAOBAB database http://baobab.sedoo.fr/DACCIWA/News/. These data are the mass concentrations of VOCs in ambient air and at emission sources. One should note that the available data are averages ± standard deviations for direct emissions. For ambient measurements, we reported the average of the two samples at each location.

Related, the authors need to include a much more robust description of how the VOC measurements were calibrated, an assessment of cartridge blanks, detection limits, uncertainties, etc. Please include a section on the instrumental methodologies that includes all of these details.

Dominutti et al.: As described in the methodology section, the VOC calibration was performed by injecting a certified low-ppb level VOC standard from NPL (National Physical Laboratory) on pre-conditioned tubes and later analysed with the same method used for the samples. Calibrated tubes were analysed before and after the analysis of the samples. Regarding blank tubes, several blanks were used during field campaigns in order to control the stability of the sorbent materials and the absence of any contamination during the sampling phase for target species Blanks usually show concentrations lower than the limit of detection. The performances of both analytical techniques deployed by LaMP and SAGE is described in the new revised version of the manuscript in the Material and Method section (line 225-227). Note that additional details about the method and quality control parameters are described elsewhere (Detournay et al., 2011, Ait Helal 2014, Keita et al., 2018). We report here the performances reported in those references (Table 1 and Table 2).

Table 1. Limit of detection (LOD) and global uncertainty (U) associated to the VOC species measured by Carbopack tubes and analysed by GC-FID techniques.

| | LOD (pptv) | U (%) | | LOD (pptv) | U (%) |
|---|---|---|---|---|---|
| hexanal | 3 | 21 | octane | 1 | n.d. |
| heptanal | 4 | 21.2 | nonane | 1 | 9.4 |
| octanal | 2 | 21.4 | decane | 1 | 5.3 |
| nonanal | 4 | 28.5 | undecane | 1 | 4.8 |
| decanal | 2 | 29.3 | dodecane | 1 | 4.9 |
| undecanal | 2 | 21.4 | tridecane | 1 | 32.6 |
| benzaldehyde | 2 | n.d. | tetradecane | <1 | 11.5 |
| benzene | 1 | 5.0 | pentadecane | 4 | 10.9 |
| toluene | 2 | 5.4 | hexadecane | 13 | 17.3 |
| ethylbenzene | 1 | 4.0 | α-pinene | <1 | 4.4 |
| m+p-xylene | <1 | 5.7 | camphene | 1 | 5.1 |
| o-xylene | 1 | 3.7 | β-pinene | 3 | 8.4 |
| styrene | 1 | n.d. | α-terpinene | <1 | n.d. |
| 1,3,5-trimethylbenzene | 1 | n.d. | limonene | 1 | 15.2 |
| 1,2,4-trimethylbenzene | 1 | n.d. | γ-terpinene | 3 | n.d. |
| 1,2,3-trimethylbenzene | <1 | n.d. | 3-carene | 1 | n.d. |
| pentane | 2 | n.d. | α-ocimene | 1 | n.d. |
| 2-methyl pentane | 1 | n.d. | terpinolene | 1 | n.d. |
| hexane | 1 | n.d. | myrcene | 2 | n.d. |

| | | | | |
|---|---|---|---|---|
| **iso-octane** | 1 | n.d. | **nopinone** | <1 | n.d. |
| **heptane** | 1 | n.d. | | | |

Table 2. Limit of detection (LOD) and global uncertainty (U) associated to the VOC species measured by Tenax tubes and analysed by GC-MS techniques.

| | LOD (pptv) | LOQ (pptv) | U (%) |
|---|---|---|---|
| **Benzene** | 10.32 | 51.59 | 14.40 |
| **Toluene** | 1.76 | 8.81 | 5.60 |
| **o-xylene** | 2.92 | 14.61 | 6.55 |
| **m+p-xylene** | 1.29 | 6.43 | 5.88 |
| **Ethylbenzene** | 3.96 | 19.78 | 6.17 |
| **Octane** | 4.01 | 20.05 | 7.49 |
| **iso-octane** | 14.58 | 72.90 | 20.20 |
| **Heptane** | 2.34 | 11.70 | 10.80 |
| **2- methylpentane** | 22.57 | 112.84 | 12.00 |
| **Isoprene** | 12.19 | 60.96 | 20.00 |
| **Limonene** | 1.17 | 5.84 | 10.90 |
| **β-Pinene** | 3.00 | 15.01 | 8.23 |
| **α-Pinene** | 8.67 | 43.35 | 19.19 |
| **1,2,3-trimethylbenzene** | 1.10 | 5.49 | 15.18 |
| **1,2,4-trimethylbenzene** | 2.25 | 11.25 | 8.54 |
| **1,3,5-trimethylbenzene** | 2.01 | 10.03 | 38.27 |

Lines 228-229: Carbonyls are ketones, so this sentence doesn't really make sense. Also, perhaps just list the "6 VOCs of intermediate volatility" to explain what is meant.

> Dominutti et al.: we have included more details about this classification in the manuscript and also change carbonyls by aldehydes (lines 233-237) : […] *VOCs can be classified according to their saturation concentration, C\*, which indicates their volatility (Ait Helal et al., 2014, Robinson et al., 2007; Epstein et al., 2010). Therefore, $C_{12}$–$C_{16}$ alkanes are classified as VOC of intermediate volatility, since given their C\* values are between 103 µgm$^{-3}$ <C\*< 106 µgm$^{-3}$ [...].*

Lines 257-269 – there are three references to Keita et al. (2018) in this paragraph alone. The strong dependence on, and inadvertent co-opting of the work that was done in the previous work to this work makes it questionable where the line between the two papers really lies. The authors should consider what is really necessary to include in this paper, and what is just additional hyperbole that was already covered in the Keita paper

> Dominutti et al.: As was stated in the previous revision, our study integrates the complete VOC database of 56 compounds instead of the only fifteen ones analysed in Keita et al 2018. Moreover, differences are not only related to the VOC species used in the emission factor calculations but also emission ratios, reactivity, ozone potential formation and secondary organic aerosol formation analyses have been addressed in this new work. Since our study have used the same calculation method as in Keita et al, we considered that this reference needs to be provided in order to avoid a copy- paste of calculation method.

Lines 347-349 – this was moved from elsewhere, but the reason for including this sentence needs to be given. I.e., there needs to be a point given for this reference info.

Dominutti et al., we agree with the reviewer and this sentence and references was moved to the next section, lines 396-400.

Lines 373-375 – were the observations reported in Djossou et al. (2018) measured at the exact same locations, or just similar types of locations?

Dominutti et al.: Despite that less sampling points were monitored in Abidjan by Djoussou et al (2018) work, two of these are the same sampling sites than in our study: such as AD and AT and one similar to our KSI site.

Lines 375-377 – this sentence is very hand-wavy. It should either be more quantitative, or removed.

Dominutti et al.: we provide new details after this sentence for a better understanding (lines 376-378)

*[...]Besides the dilution processes, the spatial distribution of total VOC concentrations seems to be related to the proximity of emission sources, affecting ambient VOC concentrations in the different sampling locations. For example, higher total VOC concentrations were mainly observed in the central urban area (like KSI, CRE and PL) where the density of emission sources is higher [...]*

Line 378: The authors report that the m+p-xylene and toluene contributions to ambient BTEX range from 9-27% and 8-31%, respectively, but the m+p-xylene and toluene components of the pie charts shown in Figure 3 appear to never be less than 25%. This needs to be clarified.

Dominutti et al.: the way the contribution of BTEX are reported in pie charts and discussed is confusing and have been modified. Indeed, the discussion in the text was based on the contribution of each BTEX to the total VOC load which is different from the contribution represented in the pie charts (related only to the BTEX total concentration). The new Figure 3 reports now the BTEX contribution as is discussed in the text. For that, we have also integrated other VOC contribution in the pie charts.

Lines 379-381: These sentences are inconsistent. The authors explain that the heterogeneity in the VOC spatial distribution could be related to the main activities (which is not specific enough – please describe) that are involved in the emission of these compounds. This needs to be backed up and/or clarified. The authors go on to say that "except for higher benzene concentrations observed in some sampling locations, the BTEX profile is rather consistent.", which seems to be contradictory to the previous sentence. Are the profiles consistent, or not? Do the distributions relate to the emissions, or not? And what about chemical loss processes? Do those have any bearing on the chemical profile in some of the sampling locations?

Dominutti et al.: we agree that both sentences brings some kind of inconsistency. We propose the new following discussion (lines 380-389) : [...]*BTEX composition is consistent*

*between PL, CRE, BIN and KSI sites with high VOC loads while an enrichment in benzene concentrations is observed at FAC, ABO, ZYOP, AT and AD sites (from 16% to 30% contribution). The BTEX composition can be affected by emissions and chemistry. The toluene-to-benzene ratio is a useful indicator of either traffic and non-traffic source or chemistry effects. On the one hand, the toluene-to-benzene at PL, CRE, BIN and KSI sites is higher than 4 which suggests the influence of sources other than traffic like industrial sources. On the other hand, the toluene-to-benzene lie between 0.8 and 1.9 at lower VOC load sites (FAC, ABO, ZYOP, AT and AD). These values are closer to the one usually observed at traffic emissions. There is no visible effect of chemistry here especially on higher aromatics like C8-aromatics with a shorter lifetime whose contribution stays almost constant regardless of the site [...]*

Lines 392-393: The authors note that similarities in VOC distributions in Abidjan are similar to other mid-latitude megacities, suggesting emissions from fossil fuel combustions for alkanes and aromatics could dominate other regional sources. There is a disconnect in the logic here – alkanes and aromatics in similar profile ◊ fossil fuel combustion. While this may be legit, the authors need to tie the observations to the emissions, either using literature values or the co-measured emissions.

Dominutti et al.: indeed, from Figure 4 the comparison shows that the distribution (or composition) of VOC in urban atmosphere is quite similar despite differences in absolute levels. This is consistent with previous studies comparing different database worldwide which did not include an African city like Abidjan (Von Schneidemesser et al., 2011; Borbon et al., 2001; Dominutti et al., 2016). This suggests that hydrocarbons are controlled by processes of same relative intensity. Observed concentrations of hydrocarbons result from primary emissions, chemical processing and dilution in the atmosphere. Dilution affects equally all the compounds by decreasing absolute levels without altering their composition. Chemistry can be neglected because the transport time between major urban sources and receptor sites is usually less than the compound lifetimes (here the shortest lifetime for trimethylbenzene is 4.3h). Finally, only emissions are expected to significantly alter the hydrocarbon composition. However, the composition is the same regardless of the location. Such commonality suggests that the urban hydrocarbon composition worldwide is controlled by emissions from fossil fuel combustion and gasoline powered vehicle in particular (see also next section).
Finally the ambient hydrocarbon distributions in Abidjan are noticeably similar to other northern mid-latitude megacities, suggesting that emissions from fossil fuel combustion for alkanes and aromatics dominate over other regional-specific sources. Even if emissions can be different in intensity (number of vehicles for instance), the hydrocarbon composition seems to be is similar. Part of this discussion have been added to the manuscript (line 403-415).

Lines 441-442 – the implication that there are terpenes in road transportation emissions is a pretty big stretch, when they are not usually found in road transportation fuels. While they may be present in waste burning, their strongest sources tend to be from biomass emissions (burning or not), and the presence of them in the vicinity road transportation emissions does not mean that they are being emitted from road transportation activities. I recommend the authors be very cautions with this kind of suggestion that isn't backed up by direct emissions studies. Also, VOC22 is not the only group that contains OVOCs, so this should be more specific.

Dominutti et al.: The discussion here is related to the VOC observed in direct emission sources and not in ambient air. The presence of terpenes was observed in biomass burning emission but also from the exhaust emission from different vehicles. Given the high concentrations observed at exhaust experiment compared to waste burning experiments during our experiments (see database at http://baobab.sedoo.fr/DACCIWA/News/.) the emissions of terpenes by cars makes sense and is not expected to be affected by other emission sources. Note that the presence of anthropogenic terpenes in urban atmospheres was already pointed out by Hellen and co-workers (Hellen et al., 2012). In that work they have found that the diurnal winter profile of these species followed the diurnal pattern of traffic, suggesting the emission of terpenes by traffic related sources. While there are very few studies, the presence of terpenes from all types of combustion sources cannot be excluded.

Lines 447-455 – There are several assumptions being made to connect and compare the OVOC measurements made in this paper to the work in previous studies. I caution the authors to not include comparisons of OVOCs that don't even include methanol, formaldehyde, acetaldehyde, acetone, etc., which are massive components of many emissions inventories. While the authors include a caveat at the end of this paragraph, I don't think that there is any need to include these details, as the limited number of OVOCs measured in their technique cannot be justifiably compared to previous studies that include C1-C4 OVOCs.

Dominutti et al.: it is absolutely true. Indeed, the OVOC measurement by off-line tubes does not include the most volatile fraction which usually represents a major fraction of OVOC. Given this limitation, these new results from OVOC are important as they represent the lower contribution limit. As a consequence we keep the VOC22 and VOC23 classification with a special footnote detailing the carbon range of the target OVOC. A sentence has been also added in the text (lines 450-452).

Technical notes:

Dominutti et al.: technical notes suggestions have been considered, and corrections have been introduced in the manuscript and figures.

Line 30: change to "despite high VOC concentrations near-source…"
Line 33: change "achieved at" to "for"
Line 37: "the VOC atmospheric impacts"
Line 42: Gg yr^-1, Gg year-1, Gg per year… be consistent throughout.
Line 44: "the residential sector is largely underestimated in global emission…"
Line 47: perhaps "for the entire West Africa region…"
Line 49: sometimes "fuelwood" is used, sometimes "fuel wood". Be consistent. Ideally, "wood fuel" would be better, if the reference is to the burning of wood for fuel.
Line 49: add "burning" or "burning and fabrication" after charcoal to be clear.
Line 58: "will continue to increase…" (they're already increasing.)
Line 60: "urban centers in the region is a significant issue" – unless you're going to list others, "one of the main issues" doesn't have any context.
Line 62: Eliminate "consequently".
Lines 64-65: "Saraha Desert, biomass burning, and local urban…"
Line 69: "Regional biomass burning is a significant source…"?
Line 72: Perhaps change "important" to "major". Important implies that the emissions are

useful or good.
Line 77: Add a comma after biofuels.
Line 79: "However, emission estimates are uncertain…"
Line 84: change "their emissions…" to "VOC emissions…"
Lines 85-86: eliminate "As VOCs are important pollutants". This is redundant, and again, implies that they're "good".
Line 87: "secondary product formation."
Line 89: "characterizing VOC to better understand their emission…"
Line 91: "VOC observations have been intensely used…"
Line 95: "For northern mid-latitude cities, discrepancies up to a factor of 10 for VOC emissions have been observed (Borgon…"
Line 99: "involve numerous…"
Line 110: "suitable data, the uncertainties…"
Line 119: "off-line" isn't necessary here.
Line 123: "…in real-condition operation" is awkward.
Line 125: add a comma after fabrication
Line 126: "additional VOC emissions"? (i.e., the ones not reported in Keita et al.?)
Lines 136-137: be consistent with the Knippertz paper with "Workpackage" vs. work package, and capitalize "Air Pollution and Health".
Line 140: "20% of the population of the…"

Lines 145, 147, 149, 343: northeasterly, southwesterly, southwesterlies – no hyphen needed.
Line 156 and elsewhere – for C6-C9 and C5-C16 - the numbers should be subscripted.
Line 160: "for 5 hours at 320 °C"
Line 162-163: "They include regional (?) background sites as well as areas impacted…" (Background can mean different things depending on the context.)
Line 167: "100 mL sccm flow rate" is redundant. Delete mL and define sccm.
Line 171: remove all hyphens from this line. Also, consider "wood fuel burning".
Line 122: "for 15 min".
Line 213: remove the period from 40° s-1
Line 240: "the mathematical basis" may be more appropriate.
Line 249: there is a + missing from the denominator of Delta-VOC/(Delta-CO + Delta CO2).
Line 251: should emission and background be subscripted?
Line 255: change the semicolon to a comma.
Line 276: There is a Delta symbol missing from the numerator.
Line 282: "et al."
Lines 284 and 295: The ° symbol should be consistent throughout the paper.
Line 292 and throughout (including the Supplement) Usually, kOH has a subscripted OH.
Line 294 and elsewhere: Spell out molecule. "molec" is not an acceptable shortform.
Line 304: Can VOC – ozone formation potential be written a way that doesn't look like "VOC minus ozone formation potential"? Similar question for line 317.
Line 306: "Derwent and co-authors (Derwent et al., 2007; 2010a; Jenkin…" – not all the papers listed are by "Derwent et al."
Line 325: it is not clear what "Table S1, in the 1" means.
Line 337: "… analysed for 2016. Meteological…" (The fact that the field campaigns were in 2016 should be included in the methods, and doesn't need to be repeated here.)
Line 344: delete the period in m s-1.
Line 352: "Keita et al. (2018)"
Line 353 "8 aromatic hydrocarbons"
Line 354: delete the space after fossil.

Lines 359-360: Delete the sentence "Details on sampling locations can be found in Table 2." This information is not needed.

Line 367: Delete "(Figure 3.)" The details stated in this sentence are not shown in Figure 3, which shows the relative contributions of the different BTEX components, and the total VOC concentrations.

Line 369: "at each site, and the BTEX…"

Line 373: "maximum aerosol concentrations".

Line 374: delete "Figure 1" from the two sites. This information isn't needed.

Line 375: "Besides the dilution process…"

Line 384: "with"; delete "blue points in "

Lines 391-392: "Our results indicate that ambient VOC distributions in Abidjan are noticeably similar to other northern…"

Line 397: "Figure 5 shows…"

Line 404: "x 10-15 cm… shouldn't be in parentheses.

Line 424: again – ketones are carbonyls. Rephrase.

Line 426: "according to the chemical function".

Line 430: "as already described".

Line 438: Section should be capitalized. And "reported" should be changed to "comprised".

Line 439: Important is the wrong word. Most significant, perhaps? Also, with regards to what?

Line 457: "charcoal burning" (to differentiate from charcoal fabrication).

Line 470: "described in Section 2.3."

Lines 479-481: This is still not a complete sentence.

Line 493: "calculated for each source"

Line 493 and Figure 7 – for numbers > 9999, either add a space between the 1000s and the 100s place, or a comma. (80343 ◊ 80,343 or 80 343).

Line 499: "charcoal burning has…"

Line 500: "SOAP values…"

Line 545: "accounting for 62%" of what?

Line 548: "for atmospheric chemistry".

Lines 550-552 – this is not a complete sentence.

Line 571: "Figure 10a shows…"

Line 575: "terpene emissions"

Line 577: controlled is not the right word.

Line 579: another example of confusing the results presented in Keita et al. with this work. "In our study" vs. "in that study"…

Lines 587-589 – this is the third time the authors have stated this. Once is enough.

Line 589 – ruled is not the right word.

Line 602 – "World Health Organization" is its official English name.

Lines 597-609 – I believe all references here to Figure 11 (a-c) should be Figure 10.

Line 620 – probably best to use "local burning" rather than "biomass burning", which usually implies widespread fires. Or just be specific about the kinds of burning that were quantified.

Line 643 – delete "to the annual emissions." (redundant with Gg year-1).

Figure 2 – the caption states that weekly precipitation is in mm/month, but the x-axis just states mm.

Figure 4 – South Africa is not a city. In the caption, please include the city and country where the measurements were made for all referenced studies.

Figure 6 – Capitalize Table S1 and Table S2.

Figure 7 – it is unclear why there are asterisks after SOAP in the (m) – (p) panels.

Figure 8 – the bottom x-axis should be fixed – "% weighted by kOH" with OH subscripted if possible.

Figure 10 – it is a little sketchy to include aldehydes here, when the number of volatile aldehydes measured is actually pretty small, and didn't include any C1-C5 aldehydes. If the authors insist on including 10c, they should at least clearly specify which aldehydes are included, i.e., C6+ aldehydes.

Dominutti et al.: we have included the clarification about aldehydes total emissions. Also expressed in lines 450-452.

Figure 11 – this figure should have error bars showing the uncertainties and/or the range of molar masses observed from the different sources.

Dominutti et al.: Figure 11 shows the molar mass contribution (%) of monoterpenes species and isoprene in the emission sources measured in Abidjan. As this figure shows the relative values, error bars are not easy to be represented. However, we improve the data representation for a better understanding.

Table 2 – AT – What kind of "transport" station? That is too vague. Also, "ancient" is not the right word – it implies centuries-old relics, not decades-old. AD – "Landfill waste burning". CRE - "much wind" could be amended to something better. "windy" for instance. ZI YOP – "all types of industries…" KSI – delete space after "traffic/"

Table S1 and S2 – the authors are encouraged to check the style guide, but generally, lower case letters are used as superscripts for footnote references, not numbers. Also, define "n.d.".

**Report #2**

Suggestions for revision or reasons for rejection (will be published if the paper is accepted for final publication)

Dominutti et al.: We thank the reviewer for these suggestions. The technical notes were considered, and corrections have been introduced in the manuscript.

Line 30 "high VOC concentrations…" reads better

Line 36 "…contribution to VOC emissions…' is better

Line 37 there is a period missing after "magnitude"

Line 37 "…govern the VOC atmospheric impacts…" or "…govern the VOC's atmospheric impacts are both better options.

Line 50 "…sources of VOC emissions…" or "…sources of VOCs" are better options.

Line 95 "For the northern mid-latitude cities alone…" sounds better

Line 121 "…VOC emissions in regional atmospheric chemistry…" or "…VOC emissions on regional atmospheric chemistry…" are better options here.

Line 375 "…Besides the dilution processes…" is better.

Line 575 "…the distribution of terpene species…" sounds better.

Line 576 "While terpene emissions from road transport…" sounds better.

Line 633 "Since few alkene species were quantified…" is more accurate.

Line 636 "…and the alkane species can…" sounds better.